# Sustainable Conservation Tillage Technique for Improving Soil Health by Enhancing Soil Physicochemical Quality Indicators under Wheat Mono-Cropping System Conditions

**Mahran Sadiq** [1,2], **Guang Li** [1,*], **Nasir Rahim** [2] **and Majid Mahmood Tahir** [2]

[1] College of Forestry, Gansu Agricultural University, Lanzhou 730070, China; khanmahran420@gmail.com
[2] Department of Soil and Environmental Sciences, University of Poonch Rawalakot, Rawalakot 12350, Pakistan; nasirrahim@upr.edu.pk (N.R.); majidmahmood@upr.edu.pk (M.M.T.)
[*] Correspondence: lig@gsau.edu.cn

**Abstract:** An improved understanding of the effect of conservation tillage on soil physicochemical quality indicators is obligatory to manage and conserve soil in a climate change scenario. Tillage strategies change soil physicochemical characteristics, consequently modifying crop yields. Conservation tillage is generally used to improve the soil physicochemical characteristics globally. However, the impact of conservation tillage on different soil depths under wheat cultivation is not well documented. A 3-year study was conducted using a randomized complete block design (RCDB). The objective of this research was to specifically study soil physicochemical indicators (soil bulk density, porosity, hydraulic conductivity, water content, temperature, nitrogen, phosphorous, potassium, C:N ratio, pH) and (crop yield) in conventional tillage (CT), straw incorporation into the conventionally tilled soil (CTS), no-tillage (NT), and stubble-retention to the no-tilled soil (NTS) measures under wheat monocropping system across different soil layers. Averaged over 0–40 cm soil layer, the results depicted scarce differences among the tillage practices regarding soil bulk density, porosity, water content and hydraulic conductivity. CT increased soil temperature over conservation tillage systems. Overall, conservation tillage improved soil total nitrogen, available phosphorous, total potassium, C:N ratio and yield than CT, whilst it decreased soil pH. We conclude that NTS and CTS are the best strategies to enhance soil health under wheat mono-cropping system conditions.

**Keywords:** bulk density; sustainable conservation agriculture; nutrients; pore space; semi-arid region; soil health; hydraulic conductivity

## 1. Introduction

Cultivated soils in many semi-arid and dry regions, like the Dingxi Northern China zone, have low organic matter because of climatic limitations that affect net primary yield. Low fertility of soil and water deficit make these agro-ecosystems susceptible to the ongoing global climate change process and the cyclic drought events. Return on investment is indeterminate with irregular weather patterns and degraded soils, which increases the economic risks for farmers. One approach to increase crop yields whilst reducing natural resources and soil degradation in such environments is to adopt nutrient-land-crop-water management strategies that are jointly encompassed in the term conservation agriculture [1–3].

Conservation tillage strategies including no or less soil inversion, soil cover maintenance with straw-return and straw incorporation into the field have been applied globally [4]. The practice of conservation tillage of soil cultivation has gained importance because of the need for soil preservation. An improved understanding of the effect of conservation tillage measures on soil physicochemical properties is obligatory to manage and conserve soil under different soil management, soil type and climatic condition scenarios [5]. This information is most significant in semi-arid areas such as the northern China

belt. Tillage has been a fundamental component of agronomic production since the first great civilizations. Tillage is performed to incorporate crop residues into the ground and to control weeds. The aim of soil tillage in agriculture is to produce appropriate soil physical conditions for the germination of seeds and growth of plants [6]. The inappropriate soil tillage management strategies of some crop cultivation methods leave the soil prone to erosion and the intensive traffic by the machinery cause a raise in the bulk density of soil [7], which can lead to soil degradation by compaction. When this occur, it is a problem for crop cultivation [8]. In order to preserve the soil integrity, all water and crop yield strategies used in agriculture should be oriented towards resource conservation.

In arid and semi-arid regions, "land management change" is important to promote the conservation of soil and water in the agricultural system. Conservation tillage systems can mitigate the impacts of dry spells [9], improve soil physicochemical properties [10] and crop yields [2]. Compared with conventional tillage systems, conservation tillage alteration reduces the soil disturbance, frequency and intensity [11]. Regarding soil physico-chemical characteristics the conservation tillage strategy has shown a great range of results [1,5]. Accordingly, examination of the impacts of these new soil conservation measures on soil physico-chemical properties is obligatory.

Tillage practices could affect the soil physical characteristics such as soil BD and water infiltration depending on the level and intensity of soil inversion [12]. The soil physical characteristics play a vital role in determining a soil's ability to store and capture rainfall water [13]. Conservation tillage practices reduce the intensity of soil inversion however, with respect to soil bulk density different conservation tillage practices have shown a great range of results [5]. For instance, studies conducted in China and Pakistan [14–16] that examined no-tillage implementation and straw application evidenced notably decreased soil bulk density. In other studies, conducted in Latin America and Iran by [17,18] claimed that conservation tillage techniques noticeably increased soil bulk density compared with conventional tillage.

Soil management has a key role in findings of soil hydraulic conductivity, for example the intensity and type of tillage and soil spatial variability. Numerous studies have explored the impact of different tillage strategies on soil hydraulic properties. Miriti et al. [19] and Karuma et al. [20] verified that different tillage measures did not notably affect soil saturated hydraulic conductivity. Research conducted in France [21] noted that soil hydraulic conductivity increased with a CT system compared to NT practices. Conservation tillage benefits concerning to the progress in soil water content may depend on natural factors, for example, enhanced biological activity, better use of rainfall, development of roots, and the soil wetting and drying cycles. For instance, Czyz and Dexter [22] reported a contribution of reduced tillage to soil water content improvement over CT. According to Zhang et al.'s [23] results, conservation tillage significantly improved the soil water content. The different soil tillage impacts and diverse results warrant the need for further study to improve understand how soil tillage measures affect soil hydraulic conductivity and soil water content [24]. Therefore, till now, the application of conservation tillage has been an issue of debate because studies under different climatic conditions and types of soil led to unconvincing differing results [5].

Soil temperature is a vital physical factor that determines soil sustainability and crop growth and production, and also regulates the heat energy exchange between the atmosphere and soil [25]. It controls the soil bio-chemical processes that in turn affect the fertilizer efficiency, germination of seeds, crop growth, and uptake of nutrients [26]. The effect of conservation soil tillage practices on soil temperature are often contradictory because soil properties, weather conditions, and soil management techniques differ enormously [24,27]. Lu et al. [28] examined how soil conservation strategies increased soil water contents and reduced soil temperature. Tillage practices influence soil physical properties, improve soil water content and reduce thermal conductivity [29].

Conservation agriculture practices improve soil nutrients and crop yield. Conversely, there are contrasting results regarding their benefits compared to CT soil practices. Conser-

vation tillage strategies are sustainable crop production approaches sought by agriculturists globally [1,2,30] The sufficient soil nutrients, for instance nitrogen (N) phosphorous (P) and potassium (K) are good indicators of soil fertility, and they are the key to better crop growth and production [31]. According to the findings of Omara et al. [32] conservation agriculture practices significantly improve the soil nitrogen compared with CT. In the same way, Han et al. [33] declared that different soil conservation management strategies notably increased the soil nitrogen and phosphorous over CT. Zhao et al. [34] reported noticeable improvements in soil nutrients under conservation agricultural measures. Wulanningtyas et al. [2] examined how no-tillage significantly increased the soil nutrients and improved soil health.

Some studies have reported reductions in crop productivity under conservation tillage measures for numerous reasons, for instance, soil compaction which reduces infiltration of water, and can decrease crop yield under soil non-inversion conditions [35]. Brennan et al. [36] reported improvements in crop growth and grain yield using a no-tillage strategy. Moreover, conservation tillage measures, especially no-tillage practices, depend on the duration of their application [37].

Studies have mostly explored the individual impacts of conservation tillage techniques (e.g., straw incorporation, straw-return and no-tillage) on crop yields and soil physicochemical properties [2,34] and the effects on individual soil physical or chemical quality indicators [1]. However, the impacts of different conservation tillage practices (straw incorporation, straw-return and no-tillage) together and on soil physicochemical quality indicators together are not clear. It is often the different conservation tillage techniques effects and benefits that encourage adoption. These effects must, consequently, be understood and optimized. Additionally, in China, studies on the impacts of conservation tillage systems on surface soil physicochemical characteristics have been conducted [33], but the information on the impacts of conservation tillage systems on sub-surface soil physicochemical properties under wheat cultivation conditions is scanty. Consequently, this study was conducted to examine the effects of conservation tillage techniques on surface as well as sub-surface soil physicochemical properties under wheat mono-cropping system conditions. Moreover, the practice of burning of crop straws after the crop harvest is still occurs in Northern China, which leads to soil degradation, deficiencies in water and nutrient availability for plants as well as yield reductions. In the recent research, we have explored the effects of different conservation tillage techniques (straw-return, and straw incorporation) on spring wheat yield.

The research objectives were to: (a) evaluate the effects of conservation tillage strategies on soil bulk density, porosity, hydraulic conductivity, gravimetric water content, water storage, and temperature across different soil layers under wheat mono-cropping system conditions; (b) explore the effects of conservation tillage measures on soil total nitrogen, available phosphorous, total potassium, C:N ratio, and pH and crop yield. This research tested the hypothesis that soil conservation tillage provides better soil physicochemical quality and spring wheat yield and improve soil health.

## 2. Materials and Methods

### 2.1. Research Site Description

Our experimental site (35°34′53″ N, 104°38′30″ E) is in the Dingxi Research Station of Gansu Agricultural University (Gansu Province, China, Figure 1). The research site is located in the northern region of China at an altitude of 2000 m above sea level. The experimental area has an average annual temperature of 6.9 °C and an irregular rainfall distribution of 400 mm, a frost-free period of 140 days, and an annual sunshine duration of 2438 h [38]. The climatic conditions of the study site are defined as semi-arid. The summer temperature can rise above 35 °C whereas winter temperatures can fall below −22 °C. The annual mean evaporation during the study period was 1531 mm. The average monthly temperature and precipitation are provided in (Supplementary Material Figure S1). The soil of the research field belonged to the sandy loam class [39]. A comprehensive study site

description has been provided in previous studies [40,41]. Before 2015, the study field was bare. Wheat has been practiced for many years in Dingxi in northern China and the crop residues, particularly wheat straws, were always removed prior to the next crop cycle.

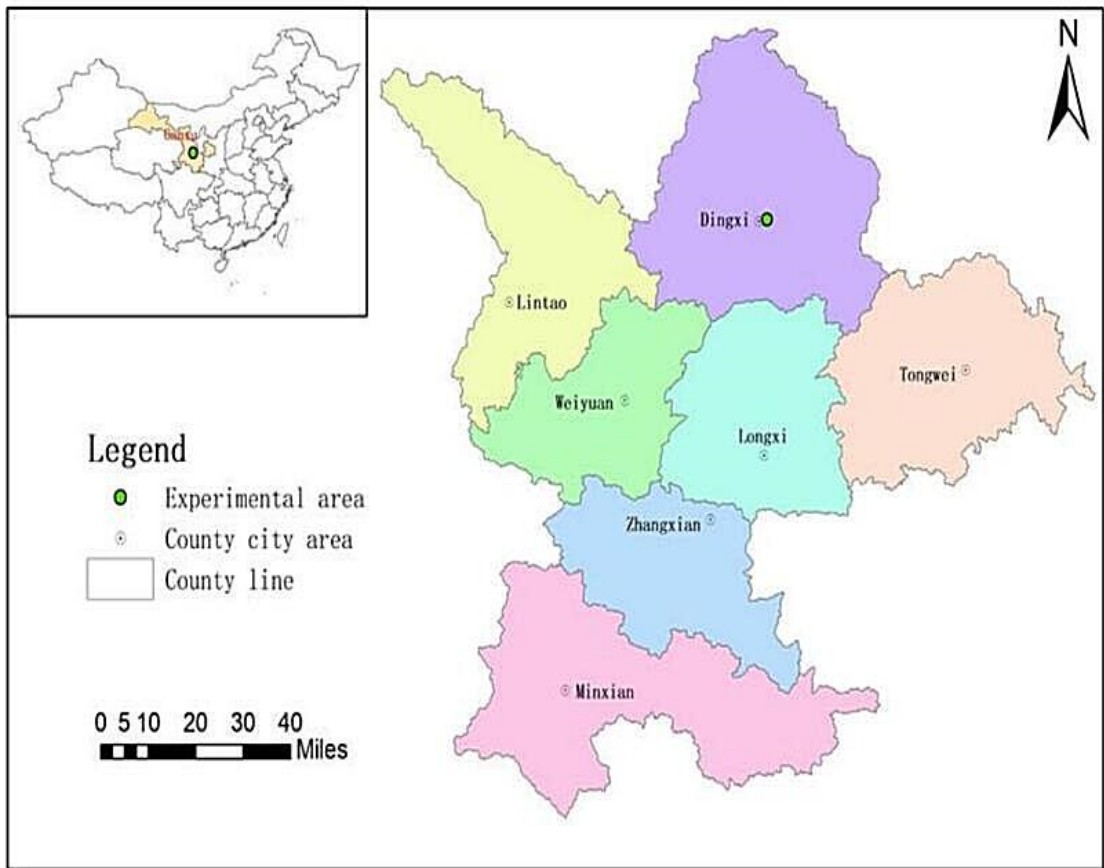

**Figure 1.** Map of study site at Dingxi Farm in Gansu Province, China.

### 2.2. Experimental Design and Setup

This research was carried out as part of an ongoing experiment originally set up in 2015 with different tillage measures (NT, CTS and CT), which were modified to include residue-return to the no-tillage system (NTS) in the subsequent years. The findings of the 3-year research project during 2017–2019 are presented in this manuscript. Crop planting was done in mid-March during the three years whilst crops were harvested in July (Table 1). The experimental study setup was a randomized complete block design (RCBD) with three blocks (B1 to B3; Figure 2). Each block involved four plots, each of 24 m$^2$. Four treatments were implemented: (1) conventional tillage (CT), (2) conventional tillage with straw incorporation (CTS), (3) no-tillage (NT) and (4) no-tillage with straw-retention (NTS). Descriptions of these tillage management practices are shown in Table 1. In the CT treatment plots, mouldboard plough at 20 cm deep ploughing was done for land cultivation followed by disc harrow and planting without crop straw. For CTS-treated plots, wheat residue after harvesting the crop followed by disc harrow and planting. In order to manage NTS treatment, wheat residue was returned to the NTS treated plots after harvesting the crop and by using no-tillage crop planter wheat was sown however, in the NT treatment wheat straws were removed after harvesting the crop and sowing was done with a no-tillage crop planter. In the 3-year study, during the crop growing seasons no irrigation was supplied; herbicide (glyphosate 30%) was applied in accordance with the manufacturer's instructions to control weeds and manual weeding was also done during the growing season when required. Crop management practices throughout the study are summarized in Table 1.

**Table 1.** Crop management practices and soil tillage treatments planning in the semi-arid Loess Plateau (China).

| Province | Station | Year | Plot Size | Crop | Crop Variety | Planting Date | Harvesting Date | Seed Rate (kg ha$^{-1}$) | Plant Density (Plant m$^{-2}$) | Fertilizer (g m$^{-2}$) | Weed Control (L ha$^{-1}$) |
|---|---|---|---|---|---|---|---|---|---|---|---|
| Gansu | Dingxi | 2017 | 24 m$^2$ | wheat | Dingxi 42 | 15 March | 20 July | 187.5 | 400 | 08 March diammonium phosphate (14.58 g/m$^2$), urea (6.25 g/m$^2$) | 10 March herbicide (Red sun) with 30% glyphosate |
| Gansu | Dingxi | 2018 | 24 m$^2$ | wheat | Dingxi 42 | 15 March | 18 July | 187.5 | 400 | 10 March diammonium phosphate (14.58 g/m$^2$), urea (6.25 g/m$^2$) | 10 March herbicide (Red sun) with 30% glyphosate |
| Gansu | Dingxi | 2019 | 24 m$^2$ | wheat | Dingxi 42 | 15 March | 15 July | 187.5 | 400 | 12 March diammonium phosphate (14.58 g/m$^2$), urea (6.25 g/m$^2$) | 10 March herbicide (Red sun) with 30% glyphosate |

| Treatment | Short forms | Description |
|---|---|---|
| $T_1$ = Conventional tillage | CT | Above-ground portions of the wheat straws were removed after the harvesting of the wheat crop. Mouldboard plough at 20 cm depth ploughing was used for cultivation of land followed by disc harrow and planting. |
| $T_2$ = Conventional tillage with crop straw | CTS | Wheat straw was homogeneously incorporated into the field after harvest of the wheat and land cultivation was performed by using mouldboard plough at 20 cm deep followed by disc harrow and planting. |
| $T_3$ = No-tillage | NT | After harvesting the wheat crop, the above-ground portions of wheat straws were removed. Wheat crop was planted in 20 cm depth by using no-tillage crop planter without using any tillage implement. |
| $T_4$ = No-tillage with crop straw | NTS | By using no-tillage crop planter in the absence of any prior tillage having wheat residue-return to the no tillage soil, wheat was sown in 20 cm deep under standing previous wheat crop stocks. |

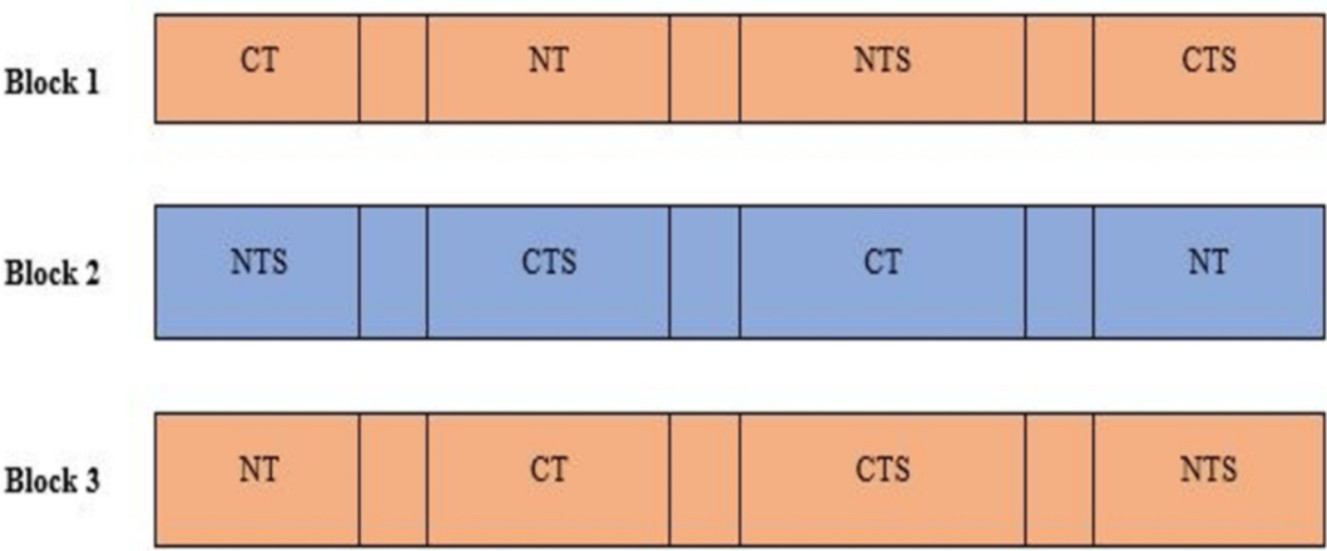

**Figure 2.** Experimental research layout and plots location for conventional and conservation tillage treatments.

### 2.3. Soil Sampling Collection and Analysis

2.3.1. Soil Physical Properties

Surface and sub-surface disturbed as well as undisturbed soil samples from CT, NT, NTS, and CTS were sampled for the determination of soil physical properties. Five soil samples were sampled from each treatment plot included different soil layers. Bulk density of soil was determined from undisturbed core samples divided into three soil depth layers (0–10 cm, 10–20 cm and 20–40 cm). The soil samples were collected by using a core sampler (steel cylinders of 5 cm diameter and 3 cm in length). The collected soil samples were processed in accordance with procedure described by [42]. It was calculated using Equation (1):

$$BD = M/V \tag{1}$$

where: B.D = soil bulk density $(g\,cm^{-3})$, M = mass of the dry soil sample (gm), V = volume of sample $(cm^{-3})$.

Soil porosity was calculated from the soil particle density and bulk density values. Using Equation (2), the percent of soil pore space was calculated:

$$P = [1 - (BD/Pd)] \times 100 \tag{2}$$

where: P = soil porosity (%), BD = soil bulk density $(g\,cm^{-3})$, Pd = soil particle density $(g\,cm^{-3})$.

Saturated soil hydraulic conductivity measurement was conducted in situ at each sub-plot to $0-40$ cm soil depth by auger-hole method, using the Guelph Permeameter. Soil hydraulic conductivity was determined by taking three steady-state readings [43]. Soil gravimetric water content was measured by taking fresh soil sample and then oven-dried at 105 °C for 72 hours and weighed [22,44]. Soil water storage was calculated from the soil gravimetric water content, soil BD, soil depth and density of water. Using equation (3), the soil water storage was calculated [44].

$$SWS = \frac{SWC \times BD \times d}{\rho w} \tag{3}$$

where: SWS = soil water storage (mm), SWC = soil gravimetric water content (%), BD = soil bulk density $(g\,cm^{-3})$, d = soil depth (cm) and $\rho w$ = density of water.

Soil temperature measurements were conducted at monthly intervals for surface and subsurface soil, 0–10 cm, 10–20 cm and 40 cm soil depths. The soil temperature was measured with a digital soil thermometer. The thermometer was based on the thermocouple principle. The thermometer output was provided in °C on the display unit.

### 2.3.2. Soil Chemical Properties

In the field, surface and sub-surface sampled five soil samples from different tillage plots were placed in a plastic bag and transported to laboratory for chemical analysis. Then, augured soil samples were dried for 6 days and sieved for chemical analysis. The soil total nitrogen was measured by using the standard semimicro–Kjeldahl digestion, distillation and titration method as described by [45]. Available soil phosphorous was measured by using the standard molybdenum antimony colorimetric method [45]. Soil potassium was measured by using the standard method [46]. The soil pH was measured by using a pH meter (Mettler-Toledo FE28, Shanghai Instruments, Shanghai, China) with a 1:2.5 soil: water ratio ($w/v$) [46].

### 2.3.3. Agronomic Traits

The wheat plant height was measured by randomly taken 10 plants per plot [47]. The dry matter yield was taken as the dry matter weight of the crop, to calculate yield per hectare. At the soil surface, a frame of (0.50 × 0.50 was placed and samples for each plant were cut to determine biological yield and then dried at 70 °C for 72 h until moisture depletion and constant weight [48].

### *2.4. Determination of Soil Health*

A Z-score test was used for determination the effect of different soil tillage management strategies on soil health [49]. Z-score formula can compute the different variables value with a definite treatment factor and also compare it to the variables mean value in all experimental treatments:

$$Zi = \frac{Xi - x}{S} \tag{4}$$

where $Zi$ = standardized value (score), $Xi$ = definite variable measured value with a factor specific experimental treatment, $x$ = mean definite variable value in all experimental treatments and $S$ = standard deviation of the certain variable in total experimental treatments.

By using Equation (4), we can explore the maximum score as a sub-total score for respective variable that was noted (BD, P, Ks, SWC, SWS, ST, TN, AP, K, C/N ratio and pH) on the basis of experimental treatment factor (different conservation tillage systems and conventional tillage). Then in order to find the general total score, each sub-total score value was added. Finally, on the basis of maximum Z-score value, we measured a comprehensive determination score from the overall total Z-score to get the best tillage system for improving or enhancing soil health.

### *2.5. Statistical Analysis*

The data obtained from the research field were tested with one-way factor interaction ANOVA at 5% probability level using a suitable computer software program SPSS 25 (IBM Corp., Chicago, IL, USA). The significant differences between different treatments and their interaction were compared with a LSD test. The relationships between different soil physico-chemical properties were analyzed using Pearson's correlation coefficient and linear regression. The data is presented as the mean values of three replications with standard deviation. Furthermore, principle component analysis (PCA) was done in order to assess the multivariate variability introduced by the different treatments for TN, AP, TK, and C:N ratio at different layers in the soil system [50].

## 3. Results

### 3.1. Soil Physicochemical Quality Indicators

Soil sampling was done in 2017, 2018 and 2019 before wheat planting in early March for measurement of soil physicochemical quality indicators. Over the 3-yr study period, the pre-sowing soil BD, porosity, SWC, SWS, TN, SOC, C/N ratio, AP, K, N-NO3-, soil temperature, pH, soil electrical conductivity under different tillage systems were 1.43 g cm$^{-3}$, 45.91%, 14.12%, 68.3 mm, 0.55 g kg$^{-1}$, 5.82 g kg$^{-1}$, 10.43, 0.37 mg kg$^{-1}$, 18.28 g kg$^{-1}$, 25.85 g kg$^{-1}$, 6.10 °C, 8.40, 0.34 dSm$^{-1}$ respectively. The measurement methods for soil physicochemical quality indicators are presented in Table 2.

**Table 2.** Physico-chemical characteristics of the pre sowing initial tested soil.

| Property | Soil Layer (cm) | | | | Measurement Method |
| --- | --- | --- | --- | --- | --- |
| | 0–10 | 10–20 | 20–40 | 40–60 | |
| Soil BD (g cm$^{-3}$) | 1.38 | 1.37 | 1.42 | 1.46 | By core sampler method |
| Soil porosity (%) | 47.92 | 48.30 | 46.41 | 44.90 | $(1 - (BD/P)) \times 100$ equation |
| Gravimetric soil water content (%) | 14.96 | 13.64 | 12.68 | 14.79 | Oven dry method |
| Soil water storage (mm) | 20.22 | 37.94 | 66.46 | 130.17 | SWC $\times$ BD $\times$ d/$\rho$w |
| Soil TN (g kg$^{-1}$) | 0.59 | 0.55 | 0.54 | 0.49 | Semimicro-Kjeldahl method |
| SOC (g kg$^{-1}$) | 5.88 | 5.92 | 5.57 | 5.68 | Walkley-Black dichromate oxidation |
| C:N ratio | 9.96 | 10.76 | 10.31 | 11.59 | SOC/TN formula |
| AP (mg kg$^1$) | 0.39 | 0.37 | 0.34 | 0.35 | Colorimetric method |
| TK (g kg$^{-1}$) | 18.41 | 18.48 | 18.34 | 18.25 | Colorimetric method |
| Soil nitrate nitrogen (g kg$^{-1}$) | 25.64 | 25.79 | 25.94 | 25.85 | 2 mol L$^{-1}$ KCl extraction |
| Soil temperature (°C) | 6.83 | 6.14 | 5.46 | 5.78 | By digital soil thermometer |
| pH | 8.40 | 8.39 | 8.41 | 8.42 | By pH meter |
| ECe (dSm$^{-1}$) | 0.33 | 0.33 | 0.36 | 0.34 | By EC meter |

Note: P.D = particle density = (2.65 g cm$^{-3}$); $\rho$w = density of water; d = soil depth.

### 3.2. Rainfall and Air Temperature

Cumulative precipitation during the years of 2017, 2018 and 2019 was 403.50 mm, 489.70 mm and 439.40 mm, respectively. The average precipitation for the three years was 444.20 mm. Consequently, water input in the first year, categorized by being particularly hot and dry, was the lowest, but precipitation was more and suitable in the second and third year. The average temperature during the three years was 7.72 °C (Supplementary Material Figure S1). Concerning the climatic conditions, 2019 could be considered the most suitable for wheat cultivation in sub-humid Northern Dingxi (China).

### 3.3. Effect of Treatments on Bulk Density in the Different Layer

Averaged across the 3-year study period, BD was significantly affected by the tillage measures for all tested soil layers (Table 3).

The maximum surface soil bulk density value (1.43 $\pm$ 0.03 g cm$^{-3}$) was measured in CT whilst the minimum soil BD was recorded in NT. Moreover, the maximum subsurface including (10–20 cm and 20–40 cm) soil BD was associated with NTS and minimum BD was noted in NT. In addition, regarding all soil layers there was no significant difference was observed in case of years (Table 3). There was no increase or decrease the BD obvious tendency for tillage in the years observed. Moreover, changes in post-harvest BD were noted for 2018 and 2019 compared to the reference post-harvest BD values in 2017. It was observed that the increase and decrease of bulk density of soil in 2018 and 2019 ranged from −2.96 to 0.70% and 2.83 to −3.81%, respectively, as shown in Table 4. In 2018, the bulk density was reduced compared with the post-harvest soil bulk density in 2017 except for NT treatment in 20–40 cm soil layer. The bulk density in 2019 increased and decreased slightly. The NTS showed greater changes compared to the other soil tillage systems. Overall, BD was increased with an increase in soil depth. The subsurface BD increased from 2.5 to 7.8% compared with the surface BD.

**Table 3.** Influence of different treatments on soil bulk density and pore space.

| | | | |
|---|---|---|---|
| **Soil Bulk Density (g cm$^{-3}$)** | | | |
| **Soil Layer Treatment** | **0–10 cm** | **10–20 cm** | **20–40 cm** |
| **Tillage** | | | |
| CT | 1.43 ± 0.03 a | 1.47 ± 0.02 ab | 1.50 ± 0.02 ab |
| CTS | 1.37 ± 0.05 b | 1.43 ± 0.04 bc | 1.46 ± 0.03 bc |
| NT | 1.33 ± 0.03 b | 1.39 ± 0.04 c | 1.42 ± 0.03 c |
| NTS | 1.42 ± 0.03 a | 1.49 ± 0.03 a | 1.52 ± 0.02 a |
| **Year** | | | |
| 2017 | 1.40 ± 0.05 a | 1.45 ± 0.05 a | 1.47 ± 0.05 a |
| 2018 | 1.37 ± 0.05 a | 1.42 ± 0.05 a | 1.46 ± 0.05 a |
| 2019 | 1.38 ± 0.06 a | 1.46 ± 0.04 a | 1.48 ± 0.03 a |
| **Soil Porosity (%)** | | | |
| Treatment | 0–10 cm | 10–20 cm | 20–40 cm |
| Tillage | | | |
| CT | 46.00 ± 1.31 b | 44.53 ± 1.03 bc | 43.36 ± 0.95 bc |
| CTS | 48.30 ± 1.91 a | 46.04 ± 1.58 ab | 44.91 ± 1.46 ab |
| NT | 49.48 ± 1.45 a | 47.25 ± 1.54 a | 46.08 ± 1.22 a |
| NTS | 45.91 ± 1.33 b | 43.73 ± 1.22 c | 42.31 ± 0.79 c |
| Year | | | |
| 2017 | 46.95 ± 2.02 a | 45.19 ± 2.01 a | 44.18 ± 1.94 a |
| 2018 | 47.77 ± 2.07 a | 46.13 ± 1.96 a | 44.47 ± 2.10 a |
| 2019 | 47.55 ± 2.35 a | 44.84 ± 1.60 a | 43.84 ± 1.47 a |

ᑫCT: conventional tillage; CTS: conventional tillage with crop straw incorporation; NT: no-tillage; NTS: no-tillage with crop straw-return. Mean values with the same letter in a column are not significantly different (Tukey 0.05).

**Table 4.** Changes of bulk density and pore space for each treatment.

| | **Increase or Decrease of Bulk Density (in Percentage) Compared with 2017** | | | | | | | |
|---|---|---|---|---|---|---|---|---|
| | **2018** | | | | **2019** | | | |
| | **CT** | **CTS** | **NT** | **NTS** | **CT** | **CTS** | **NT** | **NTS** |
| 0–10 | −1.41 | −2.96 | −2.25 | −1.40 | 0.69 | −2.20 | −3.81 | −2.12 |
| 10–20 | −2.06 | −1.43 | −1.45 | −1.36 | −0.67 | 1.39 | 2.15 | 0.67 |
| 20–40 | −0.66 | −1.38 | 0.70 | −0.66 | −1.34 | 0.68 | 2.83 | 0.65 |
| | **Increase or decrease of soil porosity (in percentage) compared with 2017** | | | | | | | |
| | **2018** | | | | **2019** | | | |
| | **CT** | **CTS** | **NT** | **NTS** | **CT** | **CTS** | **NT** | **NTS** |
| 0–10 | 1.92 | 1.04 | 2.85 | 1.09 | −0.55 | 1.31 | 2.86 | 1.38 |
| 10–20 | 2.79 | 1.63 | 1.67 | 2.32 | 1.43 | −1.67 | −1.91 | −0.85 |
| 20–40 | 0.88 | 1.69 | −0.25 | 0.3 | 1.46 | 0.83 | −2.78 | 0.90 |

CT: conventional tillage; CTS: conventional tillage with crop straw incorporation; NT: no-tillage; NTS: no-tillage with crop straw-return; negative sign indicates decrease %.

### 3.4. Effects on Soil Porosity in the Different Layer

The overview of soil porosity in accordance with tillage is presented in Table 3. Averaged over 3-year the soil porosity was notably influenced by tillage practices across various soil layers. NT significantly increased the surface as well as sub-surface pore space over other tillage systems. Moreover, no statistical difference noted for years. Maximum surface and sub-surface soil porosity was connected with 2018 whilst the minimum surface soil porosity was recorded in 2017 and subsurface soil porosity values were noted in 2019. In addition, subsurface soil porosity was decreased compared with the surface soil porosity presented in Table 3. The changes of pore space under tillage techniques in all investigated

soil layers showed a small fluctuation. The average changes in terms of increase or decrease were small for tillage systems, and ranged from −2.78 to 2.86%. The variations in soil pore space regarding different soil depths across tillage systems are shown in Table 4. The changes in 2019 were more compared with 2018.

*3.5. Effect of Treatments on Soil Hydraulic Conductivity in the Different Layer*

The analysis of variance test indicated a non-significant difference between treatments across all investigated soil layers (Table 5). Logarithmic transformed soil Ks between CT, CTS, NT and NTS amendments for 2017, 2018 and 2019 were not significantly different in any of the three soil layers increases except for the 0–10 cm and 10–20 cm soil layers in 2018. In the 3-year study, our results depicted that the highest Ks (0.51 mm h$^{-1}$) at the surface soil layer was noted with CTS in 2018, whilst the minimum Ks (0.31 mm h$^{-1}$) was recorded with NTS in 2019. Moreover, highest Ks (0.55 mm h$^{-1}$) at the sub-surface soil layer (10–20 cm) was noted with NTS in 2018. Additionally, in the 0–10 cm and 10–20 cm soil layers significant difference was noted between investigated years; the highest mean logarithmic transformed soil hydraulic conductivity (0.48 mm h$^{-1}$) was noted in 2018, whilst the lowest hydraulic conductivity was recorded in 2019. The changes of soil Ks in 2018 and 2019 under different tillage techniques in 0–10 cm soil layer, compared to the reference values in 2017 showed a great variation, being highest in NT and CTS. Compared with 2017, the Ks was increased in 2018 whilst it decreased in 2019 across all tillage systems. Our results showed significant differences in Ks by soil depth. The changes in Ks values of subsurface soil layers were highest in NTS and CT (Table 5).

**Table 5.** Effect of tillage practices on soil hydraulic conductivity.

| Soil Layer | Soil Hydraulic Conductivity (mm h$^{-1}$) | | | | | | | | |
|---|---|---|---|---|---|---|---|---|---|
| | 0–10 cm | | | 10–20 cm | | | 20–40 cm | | |
| | Year | | | | | | | | |
| Treatment | 2017 | 2018 | 2019 | 2017 | 2018 | 2019 | 2017 | 2018 | 2019 |
| CT | 0.35 a | 0.38c | 0.32 a | 0.41 a | 0.42 b | 0.44 a | 0.36 a | 0.37 a | 0.41 a |
| CTS | 0.39 a | 0.51 a | 0.34 a | 0.48 a | 0.50 ab | 0.47 a | 0.41 a | 0.40 a | 0.38 a |
| NT | 0.36 a | 0.49 ab | 0.35 a | 0.42 a | 0.44b | 0.43 a | 0.39 a | 0.41 a | 0.36 a |
| NTS | 0.38 a | 0.41 bc | 0.31 a | 0.45 a | 0.55 a | 0.46 a | 0.37 a | 0.38 a | 0.42 a |
| *p*-value | 0.08 | 0.004 | 0.65 | 0.35 | 0.008 | 0.14 | 0.57 | 0.09 | 0.66 |
| Mean | 0.37 B | 0.45 A | 0.33 C | 0.44 B | 0.48 A | 0.45 AB | 0.38 A | 0.39 A | 0.39 A |

| Soil Layer | Changes of soil Ks (in percentage) in 2018 and 2019 compared with 2017 | | | | | |
|---|---|---|---|---|---|---|
| | 0–10 cm | | 10–20 cm | | 20–40 cm | |
| | Year | | | | | |
| Treatment | 2018 | 2019 | 2018 | 2019 | 2018 | 2019 |
| CT | 8.57 | −9.37 | 2.43 | 7.31 | 2.77 | 13.88 |
| CTS | 30.7 | −14.7 | 4.16 | −2.12 | −2.50 | −7.89 |
| NT | 36.1 | −2.85 | 4.76 | 2.38 | 5.12 | −8.33 |
| NTS | 7.89 | −22.5 | 22.2 | 2.22 | 2.70 | 13.51 |

CT: conventional tillage; CTS: conventional tillage with crop straw incorporation; NT: no-tillage; NTS: no-tillage with crop straw-return. Mean values with the same lowercase letter in a column in the same year are not significantly different (Tukey 0.05) between tillage systems. Different uppercase letters represent significant differences (Tukey 0.05) between different years independently of the tillage systems; negative sign indicates decrease %.

*3.6. Effects of Treatments on Soil Gravimetric Water Content and Soil Water Storage in Early and Late Crop Growth Stages*

In the early wheat crop growth stages (March and April) averaged across the three years research, the highest SWC in the surface soil layer 11.21 ± 1.00% was associated with NTS. Minimum SWC 9.58 ± 1.09% was recorded in CT. In the 0–10 cm soil layer, compared with CT; different conservation tillage practices NTS, CTS, and NT increased by an average

of 16%, 13% and 2%, respectively, averaged across three years. In the early crop growth stages (March and April), rainfall was higher in 2018 than 2017 and 2019. Interestingly, in the subsurface 10–20 cm and 20–40 cm soil layers, the NT showed least SWC compared with other tillage practices. The subsurface soil gravimetric water content was in order CTS > NTS > CT > NT. In the late wheat crop growth stages (June and July) rainfall was lower in 2017 compared with 2018 and 2019 whilst the soil gravimetric water content was higher in 2019. Overall, after a 3-year experiment in the late crop growth stages maximum SWC in the surface soil layer $6.26 \pm 1.09\%$ was noted with NTS. CT showed the lowest at $5.43 \pm 0.27\%$ SWC. Surface SWC was in order (NTS > CTS > NT > CT). In the sub-surface soil layer 10–20 cm the CT treatment showed the least SWC compared with other tillage techniques. Moreover, in the 20–40 cm soil layer average over three years SWC was in the order: CT < CTS < NT < NTS.

In the early wheat crop growth stages averaged across 3-years the tillage had varying effects on the SWS. The straw application (CTS and NTS) increased the SWS in the investigated soil layers (Table 6). In the late crop growth stages, the surface SWS under NT, NTS, and CTS was $7.80 \pm 0.52$ mm, $8.88 \pm 1.43$ mm, and $8.56 \pm 1.23$ mm, respectively, whilst SWS under CT was $7.76 \pm 0.35$ mm. In the late crop growth stages regarding all tested soil layers, the different conservation tillage strategies notably increased soil water storage compared with CT, but a statistical significant difference was found only in the 10–20 cm and 20–40 cm soil layers.

**Table 6.** Soil water contents and water storage over the three-year period for conservation tillage systems.

| | Soil Water Content (%) | | | | | |
| | Early Crop Growth Stages | | | Late Crop Growth Stages | | |
| Soil Layer (cm) | 0–10 cm | 10–20 cm | 20–40 cm | 0–10 cm | 10–20 cm | 20–40 cm |
|---|---|---|---|---|---|---|
| | Treatments (T) | | | | | |
| CT | $9.58 \pm 1.09$ a | $9.39 \pm 1.32$ ab | $9.43 \pm 1.03$ a | $5.43 \pm 0.27$ a | $4.41 \pm 0.38$ b | $4.21 \pm 0.30$ b |
| CTS | $10.12 \pm 1.20$ a | $10.33 \pm 1.17$ a | $9.21 \pm 1.00$ a | $6.25 \pm 0.41$ a | $5.34 \pm 0.41$ a | $4.51 \pm 0.29$ ab |
| NT | $9.70 \pm 1.86$ a | $8.35 \pm 1.40$ b | $8.47 \pm 0.90$ a | $5.87 \pm 1.18$ a | $4.91 \pm 0.85$ ab | $4.72 \pm 0.62$ a |
| NTS | $11.21 \pm 1.00$ a | $9.89 \pm 1.20$ a | $9.82 \pm 1.35$ a | $6.26 \pm 1.09$ a | $5.11 \pm 0.74$ ab | $4.75 \pm 0.37$ a |
| | Years (Y) | | | | | |
| 2017 | $9.79 \pm 1.17$ a | $8.89 \pm 1.53$ a | $8.85 \pm 1.24$ a | $5.75 \pm 0.93$ a | $4.86 \pm 0.74$ a | $4.56 \pm 0.35$ a |
| 2018 | $10.18 \pm 1.64$ a | $9.93 \pm 1.19$ a | $9.50 \pm 0.98$ a | $6.03 \pm 0.85$ a | $4.92 \pm 0.66$ a | $4.61 \pm 0.58$ a |
| 2019 | $10.48 \pm 1.45$ a | $9.65 \pm 1.43$ a | $9.30 \pm 1.03$ a | $6.08 \pm 0.89$ a | $5.05 \pm 0.73$ a | $4.48 \pm 0.44$ a |
| | Soil water storage (mm) | | | | | |
| | Early crop growth stages | | | Late crop growth stages | | |
| | 0–10 cm | 10–20 cm | 20–40 cm | 0–10 cm | 10–20 cm | 20–40 cm |
| | Treatments (T) | | | | | |
| CT | $13.79 \pm 1.51$ a | $27.60 \pm 1.64$ ab | $56.20 \pm 2.78$ a | $7.76 \pm 0.35$ a | $12.96 \pm 1.16$ b | $25.26 \pm 2.13$ b |
| CTS | $13.76 \pm 2.45$ a | $29.95 \pm 1.15$ a | $54.15 \pm 2.46$ a | $8.56 \pm 1.23$ a | $15.27 \pm 2.02$ a | $26.33 \pm 1.65$ ab |
| NT | $12.70 \pm 1.22$ a | $23.71 \pm 2.89$ b | $55.12 \pm 1.23$ a | $7.80 \pm 0.52$ a | $13.64 \pm 1.25$ ab | $26.80 \pm 2.53$ ab |
| NTS | $13.60 \pm 1.19$ a | $29.67 \pm 2.32$ a | $56.45 \pm 2.34$ a | $8.88 \pm 1.43$ a | $15.22 \pm 1.54$ a | $28.88 \pm 1.92$ a |
| | Years (Y) | | | | | |
| 2017 | $13.51 \pm 2.12$ a | $25.78 \pm 1.20$ a | $51.33 \pm 0.46$ a | $8.05 \pm 0.71$ a | $14.09 \pm 0.65$ a | $26.81 \pm 1.33$ a |
| 2018 | $14.86 \pm 1.95$ a | $27.40 \pm 0.82$ a | $55.10 \pm 2.18$ a | $8.26 \pm 0.45$ a | $13.97 \pm 1.15$ a | $26.92 \pm 1.75$ a |
| 2019 | $15.50 \pm 1.58$ a | $27.98 \pm 1.72$ a | $53.94 \pm 1.91$ a | $8.39 \pm 1.12$ a | $14.74 \pm 1.77$ a | $26.52 \pm 0.86$ a |

CT: conventional tillage; CTS: conventional tillage with crop straw incorporation; NT: no-tillage; NTS: no-tillage with crop straw-return. Mean values with the same letters are not significantly different (Tukey 0.05).

### 3.7. Effects of Treatments on Soil Temperature in the Different Soil Layers

The average monthly ambient temperature varied between $-8.01$ °C and $22.62$ °C during the 2017–2019. The highest air temperature was observed in 2017 and the lowest ambient temperature was recorded in 2018. Average surface ST ranged from a minimum of a 5 °C to a maximum of 27.1 °C during the growing seasons of the three years. Soil temperature in CT and CTS was higher when the air temperature was high. The ST noted in NT and NTS was inferior in comparison with CTS and CT. Compared with conservation tillage system CT increased ST. The ST followed the trend of CT > CTS > NTS > NT (Figure 3a). Moreover, the CTS, NTS and NT treatments reduced the ST indicating the temperature moderation effect. Additionally, in the first and second year (2017 and 2018) a significant difference was noted among tillage systems but ST had no significant difference among different tillage systems for the year 2019; however, conservation tillage techniques reduced the soil temperature over CT. Furthermore, a significant difference was noted in the case of the interaction factor between years and treatments. ST was highest in 2017 over 2018 and 2019 whilst the lowest ST was recorded in 2019. In the sub-surface 10–20 cm and 20–40 cm soil layers, the ST was not significantly impacted by tillage measures for any of the investigated years; however, reducing ST trend was found under conservation tillage strategies compared with CT. In addition, the NTS, CTS and NT affected the Z-score of ST at various soil layers (Figure 4). In surface and sub-surface soil layer, CTS, NTS and NT showed the highest scores over CT.

### 3.8. Influence of Treatments on Soil Chemical Indicators in the Different Soil Layers

Our study over 3 years of soil management practices, depicted that different conservation tillage measures had a noticeable effect on soil chemical quality indicators (soil TN, available P, TK, C/N ratio and pH). In soil layers with 0–10 cm and 10–20 cm depth, NT, NTS and CTS provide significant improvements to soil TN and AP accumulation except for AP at 10–20 cm whilst no statistical difference was recorded in the case of the 20–40 cm soil layer (Table 7). The soil TK, and C/N ratio were not significantly affected under NTS, NT and CTS in any of the tested soil layers. The influence of conservation tillage technique declined with increasing soil depth, at a depth of 20–40 cm in case of TK and C:N ratio (Figure 5). Moreover, CTS, NTS and NT had no influence on pH, however, straw implementation decreased the surface soil pH (Table 8). Generally, CTS treatment gives the highest soil TN, available P, TK, and C/N ratio value compared to other treatments. The NTS treatment revealed better consequences in the improvement of soil TN, available P, TK, C/N ratio at different soil layers compared with NT. Compared with CT, conservation tillage strategies increased soil TN, available P, TK, C/N ratio. Soil TN followed the trend of CTS > NTS > NT > CT. The straw implementation (NTS and CTS) were significantly ($p < 0.05$) effective in increasing surface soil AP with CTS being the highest and NT recording the minimum with respect to NTS. Interestingly, although soil available P depicted the same trend as that of average TN, a small influence occurred with the tillage systems throughout different tested years with NTS having the maximum level during 2019 and CTS maximum during 2018 and 2019.

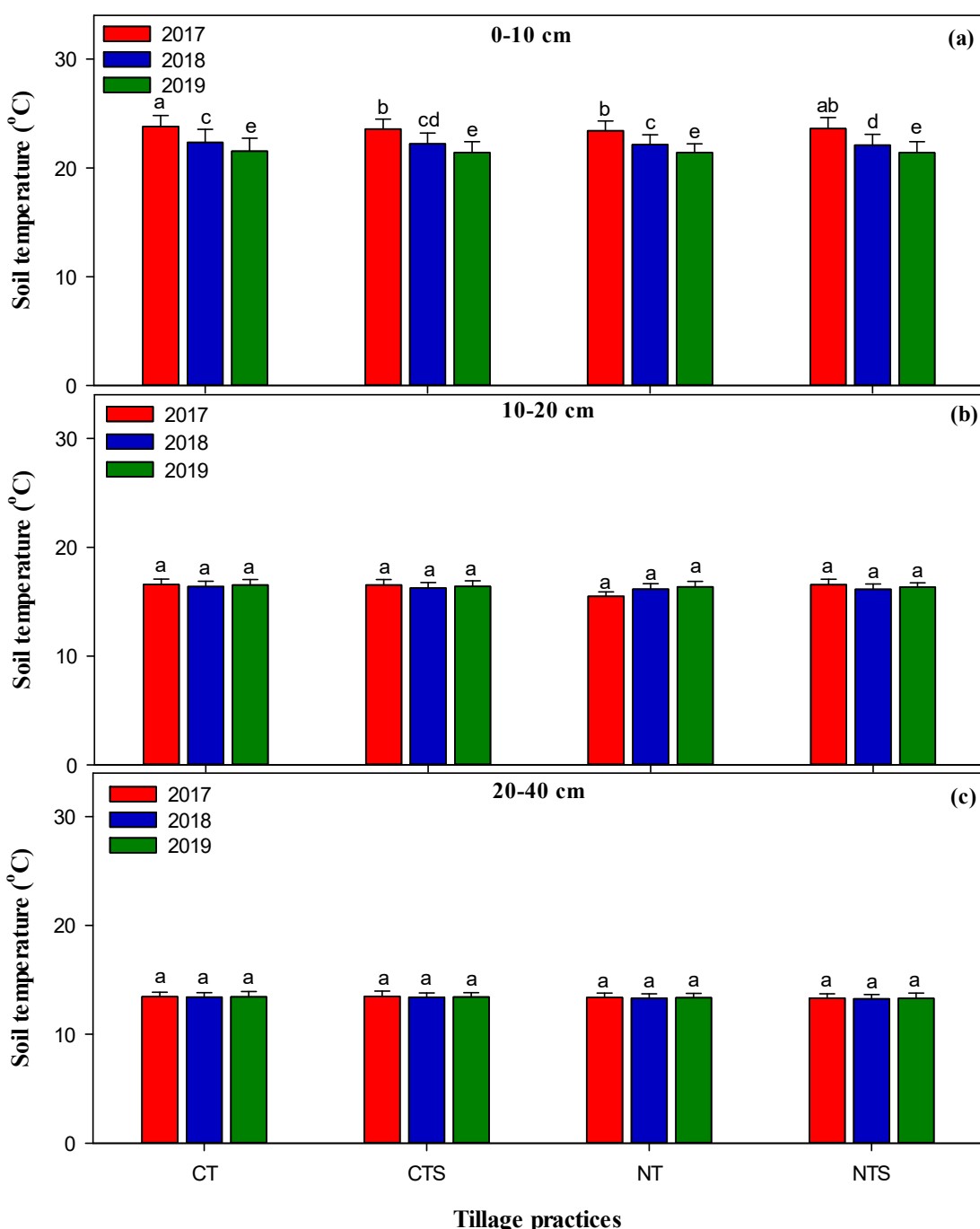

**Figure 3.** Effect of conservation tillage practices on soil temperature (°C) for the 2017, 2018, and 2019. Lowercase letters indicate least significant difference (Tukey 0.05) between treatments and years. Note: (**a**) is the soil temperature values at a depth of 0−10 cm; (**b**) is the soil temperature values at a depth of 10−20 cm; (**c**) is the soil temperature values at a depth of 20−40 cm.

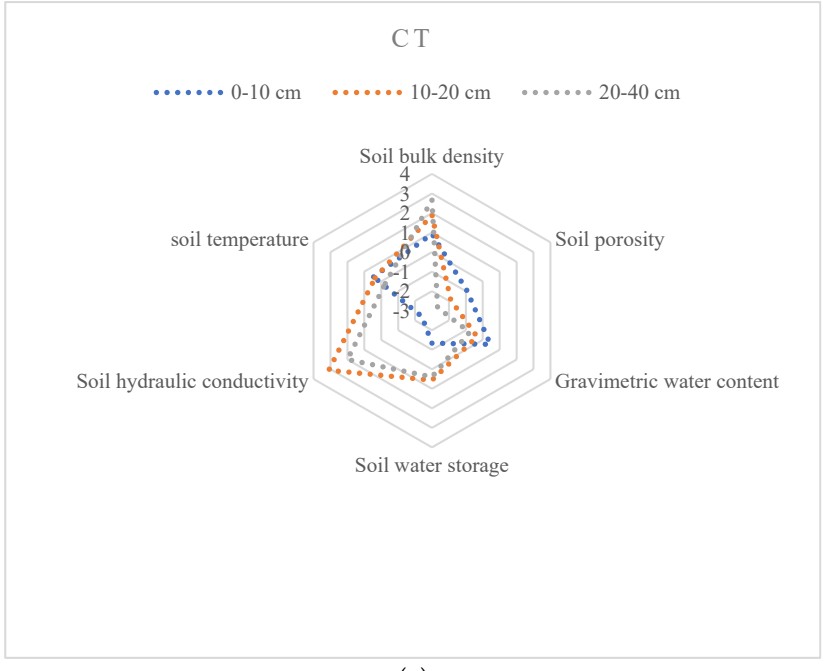

(a)

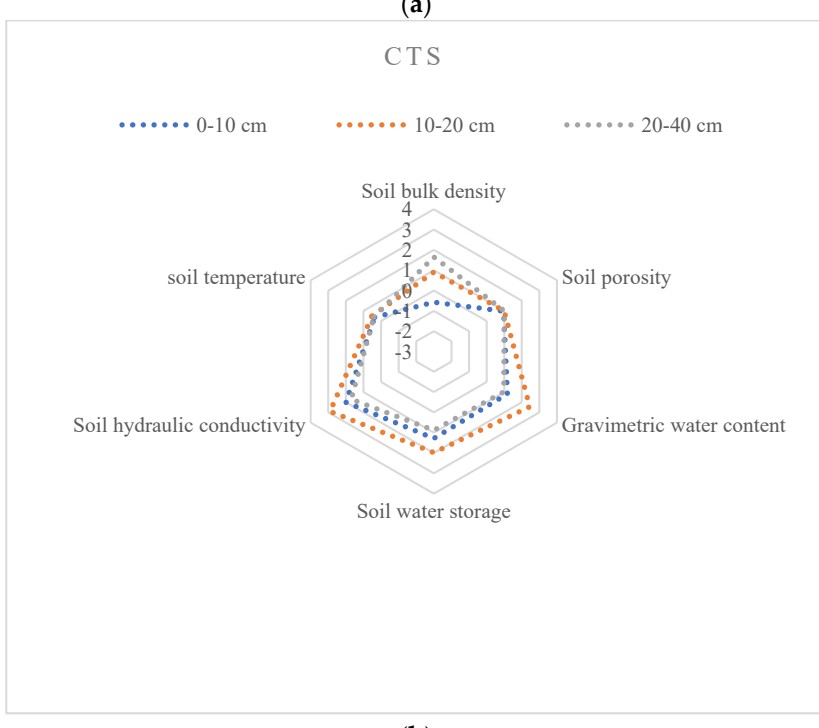

(b)

**Figure 4.** *Cont.*

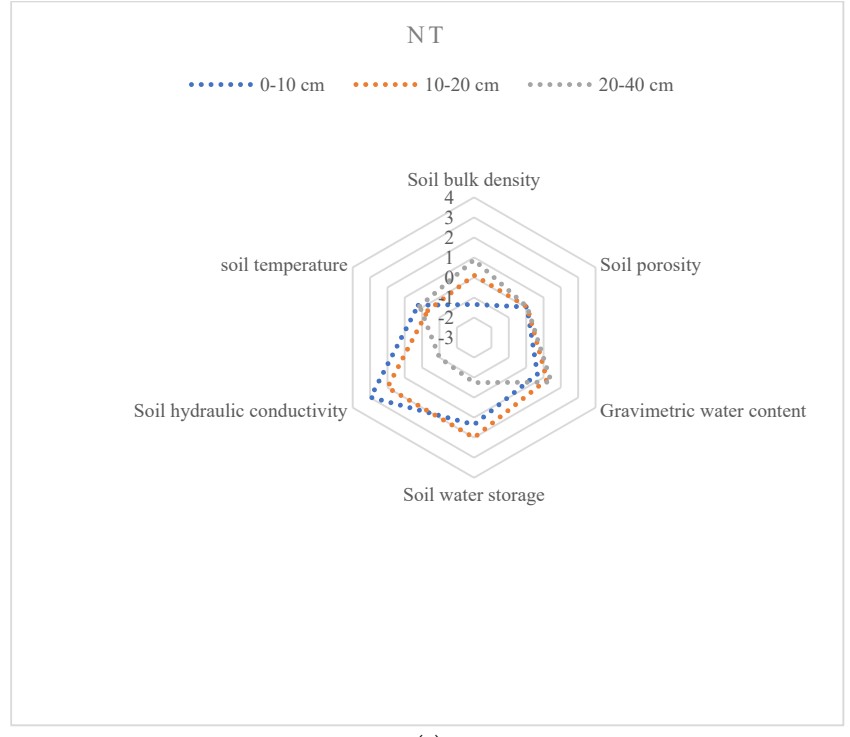

(c)

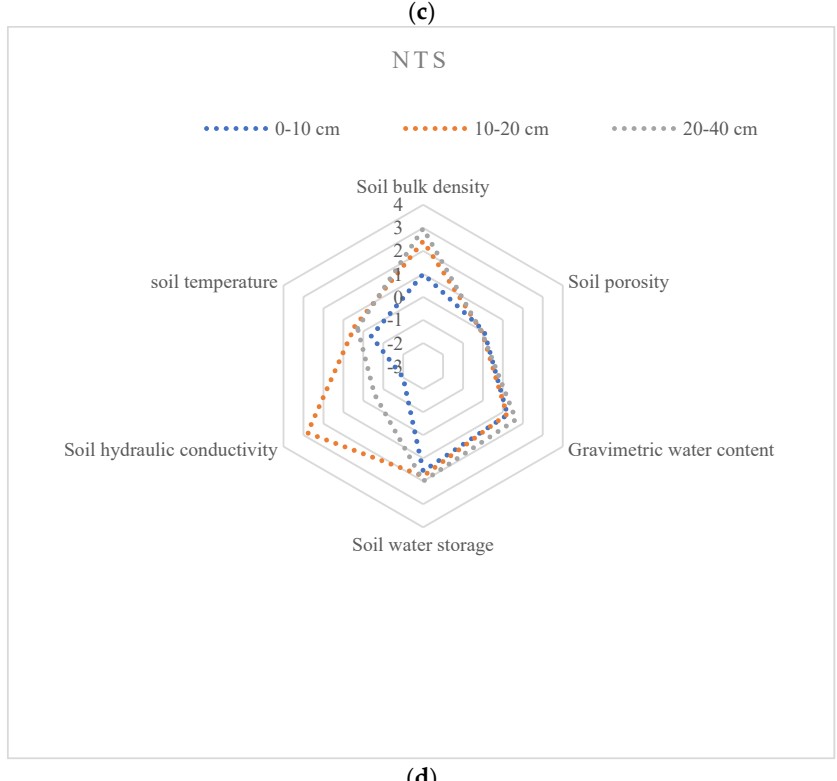

(d)

**Figure 4.** Z-score representation of the soil physical indicators for the corresponding treatments across different soil layers. Note: (**a**) is Z-score representation of the soil physical indicators for the CT treatment; (**b**) is Z-score representation of the soil physical indicators for the CTS treatment; (**c**) is Z-score representation of the soil physical indicators for the NT treatment; (**d**) is Z-score representation of the soil physical indicators for the NTS treatment.

**Table 7.** ANOVA table of soil chemical indicators in the different layer.

| | Total Nitrogen (TN) | | | | | | Available Phosphorous (AP) | | | | | | Potassium (K) | | | | | | C/N (Ratio) | | | | | |
|---|---|---|---|---|---|---|---|---|---|---|---|---|---|---|---|---|---|---|---|---|---|---|---|---|
| | | | | | | | | | | Soil Layer (cm) | | | | | | | | | | | | | | |
| Source of variation | 0–10 | p-value | 10–20 | p-value | 20–30 | p-value | 0–10 | p-value | 10–20 | p-value | 20–30 | p-value | 0–10 | p-value | 10–20 | p-value | 20–30 | p-value | 0–10 | p-value | 10–20 | p-value | 20–30 | p-value |
| Treatment (T) | * | 0.00 | * | 0.00 | * | 0.00 | * | 0.00 | * | 0.01 | * | 0.01 | n.s. | 0.22 | n.s. | 0.35 | n.s. | 0.41 | n.s. | 0.19 | n.s. | 0.26 | n.s. | 0.67 |
| Year (Y) | n.s. | 0.91 | * | 0.01 | n.s. | 0.28 | * | 0.00 | n.s. | 0.47 | n.s. | 0.25 | n.s. | 0.38 | n.s. | 0.42 | n.s. | 0.56 | n.s. | 0.39 | n.s. | 0.49 | n.s. | 0.48 |
| (Y∗T) | n.s. | 0.16 | n.s. | 0.89 | n.s. | 0.91 | n.s. | 0.77 | n.s. | 0.86 | n.s. | 0.38 | n.s. | 0.87 | n.s. | 0.88 | n.s. | 0.96 | n.s. | 0.31 | n.s. | 0.87 | (n.s.) | 0.92 |

Indicates significance at * $p < 0.05$.

**Table 8.** Soil pH under different tillage systems during 2017–2019.

| | Soil pH | | | | | | | | | | | |
|---|---|---|---|---|---|---|---|---|---|---|---|---|
| | Soil Layer | | | | | | | | | | | |
| Treatment | 0–10 cm | | | | 10–20 cm | | | | 20–40 cm | | | |
| | Year | | | | | | | | | | | |
| | 2017 | 2018 | 2019 | 2017–2019 | 2017 | 2018 | 2019 | 2017–2019 | 2017 | 2018 | 2019 | 2017–2019 |
| CT | 8.40 ± 0.02 a | 8.41 ± 0.01 a | 8.40 ± 0.01 a | 8.40 ± 0.01 A | 8.41 ± 0.02 a | 8.41 ± 0.01 a | 8.40 ± 0.01 a | 8.41 ± 0.01 A | 8.42 ± 0.02 a | 8.43 ± 0.01 a | 8.43 ± 0.01 a | 8.42 ± 0.01 A |
| CTS | 8.39 ± 0.05 a | 8.39 ± 0.04 a | 8.37 ± 0.02 a | 8.38 ± 0.03 A | 8.40 ± 0.03 a | 8.40 ± 0.02 a | 8.39 ± 0.03 a | 8.39 ± 0.02 A | 8.39 ± 0.04 a | 8.40 ± 0.03 a | 8.39 ± 0.03 a | 8.39 ± 0.03 A |
| NT | 8.40 ± 0.02 a | 8.40 ± 0.02 a | 8.38 ± 0.02 a | 8.39 ± 0.02 A | 8.41 ± 0.05 a | 8.41 ± 0.02 a | 8.40 ± 0.02 a | 8.41 ± 0.01 A | 8.41 ± 0.01 a | 8.42 ± 0.03 a | 8.40 ± 0.02 a | 8.41 ± 0.02 A |
| NTS | 8.38 ± 0.03 a | 8.41 ± 0.02 a | 8.36 ± 0.02 a | 8.38 ± 0.03 A | 8.40 ± 0.02 a | 8.41 ± 0.01 a | 8.37 ± 0.01 a | 8.39 ± 0.02 A | 8.40 ± 0.01 a | 8.42 ± 0.03 a | 8.39 ± 0.01 a | 8.40 ± 0.02 A |
| Mean | 8.39 ± 0.03 A | 8.40 ± 0.02 A | 8.38 ± 0.02 A | 8.39 ± 0.02 | 8.40 ± 0.02 A | 8.41 ± 0.01 A | 8.39 ± 0.02 A | 8.40 ± 0.02 | 8.40 ± 0.02 A | 8.42 ± 0.02 A | 8.40 ± 0.02 A | 8.41 ± 0.02 |

CT: conventional tillage; CTS: conventional tillage with crop straw incorporation; NT: no-tillage; NTS: no-tillage with crop straw-return. Mean values with the same lowercase letter in a column in the same year are not significantly different (Tukey 0.05) between tillage systems. Same uppercase letter represents non-significant difference (Tukey 0.05) between different years independently of the tillage systems.

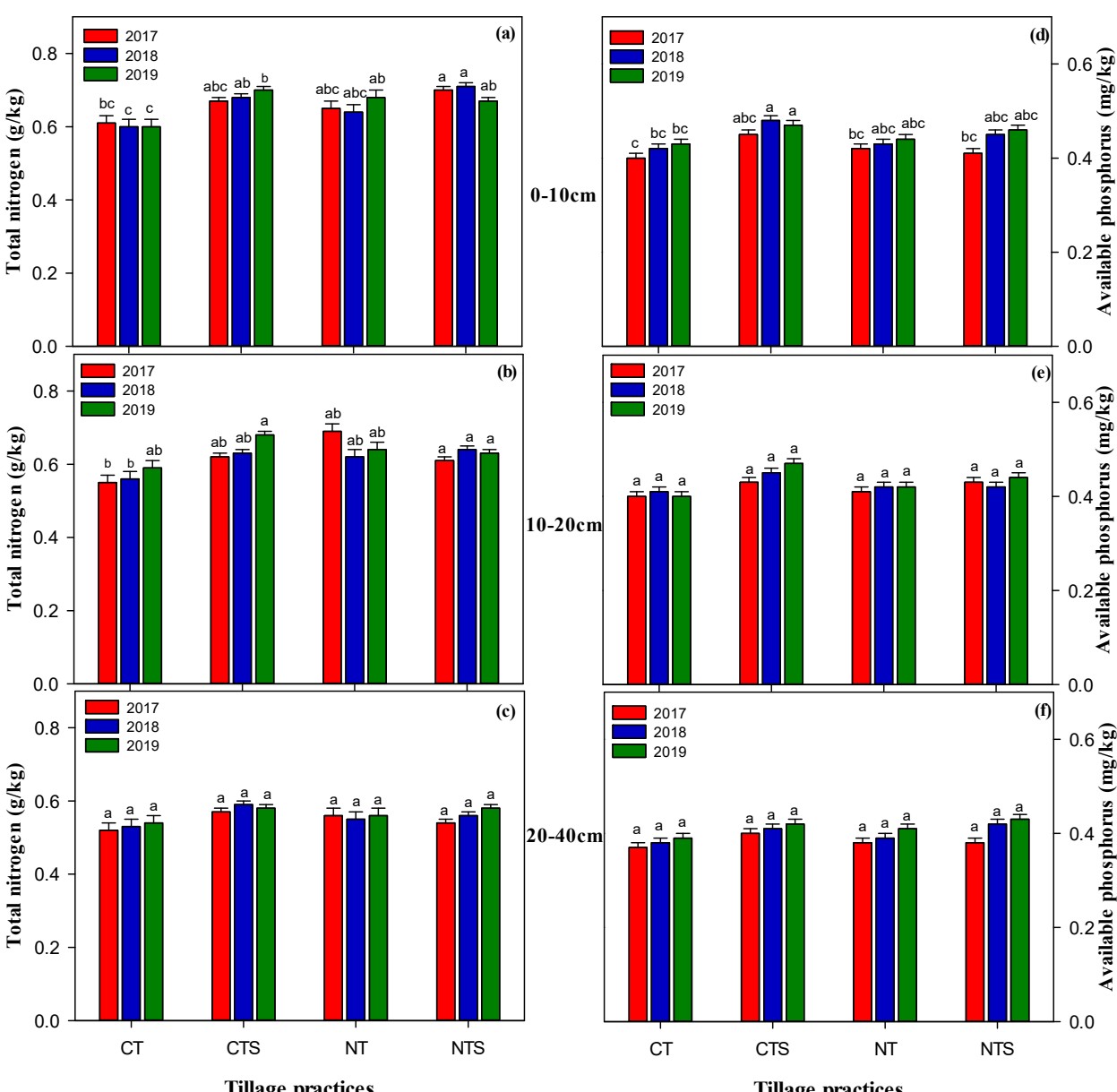

**Figure 5.** *Cont.*

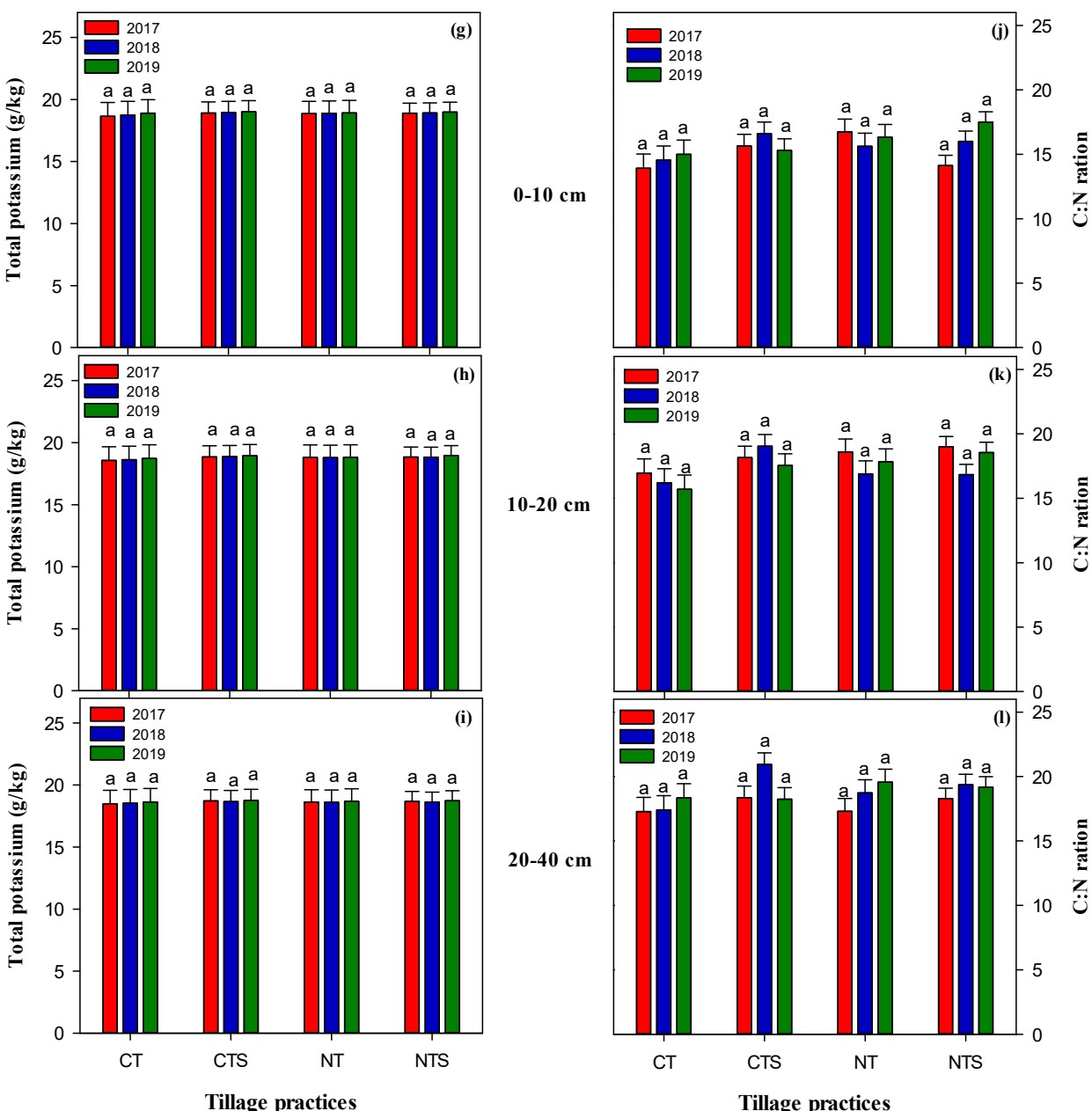

**Figure 5.** Effect of tillage strategies on soil chemical properties for the 2017, 2018, and 2019. Lowercase letters indicate least significant difference (Tukey 0.05) between treatments and years. Note: (**a**) is the concentrations of soil total nitrogen at a depth of 0−10 cm; (**b**) is the concentrations of soil total nitrogen at a depth of 10−20 cm; (**c**) is the concentrations of soil total nitrogen at a depth of 20−40 cm; (**d**) is the concentrations of soil available phosphorus at a depth of 0−10 cm; (**e**) is the concentrations of soil available phosphorus at a depth of 10−20 cm; (**f**) is the concentrations of soil available phosphorus at a depth of 20−40 cm; (**g**) is the concentrations of soil total potassium at a depth of 0−10 cm; (**h**) is the concentrations of soil total potassium at a depth of 10−20 cm; (**i**) is the concentrations of soil total potassium at a depth of 20−40 cm; (**j**) is soil C:N ratio values under different treatments at a depth of 0−10 cm; (**k**) is soil C:N ratio values under different treatments at a depth of 10−20 cm; (**l**) is soil C:N ratio values influenced by different tillage practices at a depth of 20−40 cm.

### 3.9. Effects on Crop Agronomic Traits

The tillage strategies had a notable effect on crop agronomic traits. Over 3 years of tillage practices, the different soil conservation tillage strategies altered the spring wheat plant height (Figure 6). The maximum plant height was observed under CTS whilst the minimum plant height was recorded in CT treatment; however, non-significant differences

were found between the NTS and NT. The CTS, NTS, NT improved the plant height by 6.88, 5.30, 4.82% respectively than CT. The plant height followed the trend of CT < NT < NTS < CTS.

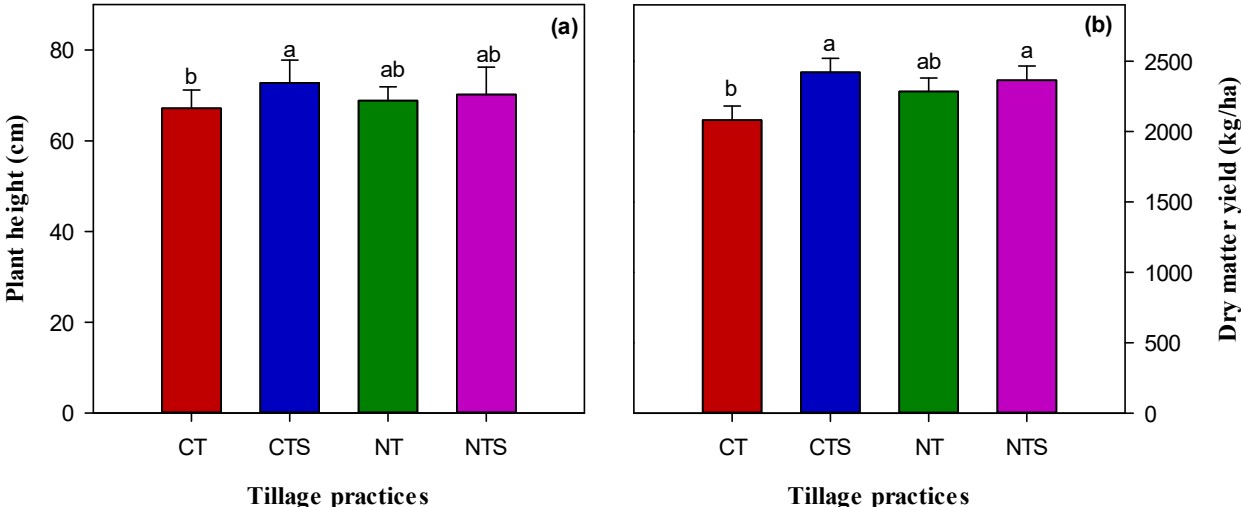

**Figure 6.** Effect of tillage measures on crop agronomic characteristics. Lower-case letters indicate the least significant difference (Tukey 0.05). Note: (**a**) is the plant height values under different tillage treatments; (**b**) is the wheat dry matter yield as affected by the different tillage techniques.

Crop dry matter yield was influenced by conservation tillage techniques and different over the investigated years (Figure 6). Averaged over the 3-year study, significant differences were noted among all treatments. Conservation tillage systems increased the dry matter yield in comparison with CT. Generally, the average percentage increase in dry matter yield for all the conservation tillage techniques (CTS, NTS, and NT) was 16.33, 13.60, 9.75% respectively compared with CT. The crop dry matter yield was in order CTS > NTS > NT > CT.

## 4. Discussion

### 4.1. Tillage Effects on Soil Bulk Density

Within the research results, NTS and CT increased soil BD compared with NT and CTS across different soil layers. Soil BD changes for all treatments were similar, and from first to third year decreased/increased slightly regarding all tested soil layers. The higher surface as well as subsurface BD under NTS was associated to the least disturbance of soil by eradicating preceding tillage, which led to soil compactness and might also be related with straw retention which led to effects of raindrops beating on soil [51,52]. The higher BD under CT was related to direct heavy machinery traffic physical compaction [53]. Interestingly, our findings also showed that BD under NT and CTS was lesser compared with CT, probably as a consequence of the sufficient organic material improvement and maybe increased biological organisms' activity [54]. The reason for the reduction in BD under CTS treatment was as a consequence of the complete inversion of the soil by the tillage implements. Likewise, organic materials addition leads to an improvement in the soil aggregation and consequently a reduction in BD. This result confirmed the findings of Higashi et al. [54] who found the conservation tillage may decrease BD. Moreover, our results are in contrast with [18] where it was declared that BD was increased under NT. Unlike the recent study, other researchers reported higher BD under NT as compared to CT [53].

The changes in terms of increase or decrease of BD during study displayed small fluctuations. Within a growing season, BD could decrease or increase because of several factors, for example rainfall intensity and volume, crop type and drying and wetting of

soil [55]. Small decreases or increases of BD have also been recorded in another studies exploring the effect of tillage systems on soil physical properties [56]. Our findings contradict the study of [57] that examined BD and reported a non-significant difference under tillage practices in different soil layers. Soil BD can vary over time, however not necessarily with a constant trend. Compared with the first year (2017), the mean subsurface BD in 2019 was slightly higher, independently of the soil tillage system. Moreover, in our experiments the subsurface BD was higher compared with the surface BD as presented in Table 4, similar to previous results [53]. The changes in terms of percentage in this study were consistent with research reported by [48] exploring the impact of tillage systems on soil physical properties. The present study variation in BD ranged from −3.81 to 2.83% compared with the first year. Similarly, our results are parallel to the findings of [58] who noted changes in BD and declared that BD varied from 1 to 11% compared with the first year.

### 4.2. Tillage Impacts on Soil Pore Space

Pore space is strongly related to the physical behavior of soil and water dynamics [59]. Soil porosity varies between different soil tillage techniques. Recently, the study of pore space under different tillage practices was one of the hotspots in soil tillage research [60]. The soil porosity changes are accordingly connected with soil bulk density values, and pore space values are the highest with conservation tillage practices, whereas the bulk density of soil decreased using conservation tillage measures [56]. Within the present study NT and CTS increased the pore space due to organic matter accumulation under conservation agriculture that led to reduce BD and increased porosity [54,56]. Moreover, during the 3-yr study regarding different soil layers; the calculated soil pore space values ranged from 45 to 50% within the normal range of soil porosity for agricultural soils [61]. Our results in this research are similar to [62] where the soil pore space values ranged from 46 to 49% after four years of study.

Generally, soil pore space was decreased with a decrease in soil depth. The subsurface soil porosity was decreased from 2.7 to 9.0% compared with the surface soil porosity (Table 3), which agrees with the results of [51,59]. Our findings are in contrast with [16] who conducted a field experiment and reported that CT increased soil pore space compared with minimum tillage practices.

### 4.3. Influences of Conservation Tillage System on Hydraulic Conductivity

Logarithmic transformed soil Ks between different tillage practices CT, CTS, NT and NTS amendments for investigated period (2017 to 2019) was not significantly different across different soil layers, except in 2018 when the tillage systems significantly affected Ks values in the cases of 0–10 cm and 10–20 cm soil layer. Moreover, the Ks values notably varied across the years, being highest in 2018 and lowest in 2019. The inconsistencies in soil hydraulic conductivity values might be due to the complexity of the variations in the soil physical environment caused by tillage operations. The average trend of Ks among treatment was CT < NT < NTS < CTS. Our results indicated non-significant differences between different tillage systems; however, CTS, NTS, and NT increased Ks over CT. Singh et al. [63] noted that the lowest soil Ks under CT practice might be attributed to the reduction of macro-porosity and destruction of aggregates in tilled soils. The positive effect of conservation tillage practices on soil Ks was not surprising, as the universal concept is that an increase in BD and compaction is stereotypically connected with least conductivity. Effective macroporosity that determines Ks are the root tubes generated by herbaceous vegetation and the aggregate faces. Ks does not depend on the absolute value of macroporosity, but on the vertical interconnection that this macroporosity presents which is broken by tillage. For this reason, although tillage can increase macroporosity, most of this macroporosity is not effective. However, NT can decrease the total macroporosity value, but it exhibits a much higher vertical interconnection. This justifies that CT presents Ks lower than NT, which is especially notable in the 0–10 cm layer, which is the most affected by soil management (tillage type: CT or NT). The organic matter accumulation

related with condensed soil physical aeration and straw accumulation could be ascribed to the improved conductivity. Accumulation of organic matter will rise the macro-porosity under conservation farming practices, thus increasing soil Ks. Our results are in line with Pikul et al. [64] who confirmed an increase in soil ks under conservation tillage practices compared with tilled soil practice, possibly because of the straw retention that modified the structure of soil, and hence aggregate stability. No significant impact of tillage on Ks over time has also been recorded in other studies [56,57]. Our results confirmed the findings of [20] who found that Ks was not notably affected by different tillage strategies. In addition, this finding contrasts with results by [19] who found significant differences between tillage systems on Ks.

### 4.4. Tillage Effects on Soil Water Content and Storage

Stored water in unsaturated soil layers is commonly known as soil water content [65]. SWC is a relation between the biosphere and the atmosphere; it affects the collaboration of the hydrological processes in the soil system over both time and space [66]. It also plays a significant role in the universal water cycle by regulating the infiltration and evaporation distribution [67]. Information of the water distribution in conservation tillage measures is essential in understanding a soil's inherent capacity to supply essential soil nutrients and the degree of the modification of soil nutrient providing capacity with tillage systems. It will offer crops optimal fertilizer recommendations for understanding sustainable production.

Within the present study, in the early crop growth stages SWC was higher compared with late crop growth stages because of high precipitation and low ambient temperature. Nevertheless, SWC was lower at late crop growth stages, mainly due to high transpiration and temperature. In addition, our results suggest that SWC under NTS and CTS was notably higher due to straw retention. Straw mainly increased the available water capacity and water infiltration [23]. The results were in agreement with [20,22] who claimed that straw application noticeably increased SWC. Overall, regarding the interaction between years and soil layers the results showed that SWC was not influenced by the different tillage techniques, however minor and inconsistent SWC variations existed between the tillage practices used in this research. Our consequences were parallel with [68] who noted non-significant differences between tillage measures in six soil layers under a maize cropping system. Generally, presence of residues preserves more water compared with CT [57]. Additionally, our results showed that higher SWC under NT treatment over CT due to the flip the earth, inherent to CT, causes a strong water loss due to evaporation. This water loss is avoided in NT, without the need to add straw. Moreover, SWS under conservation tillage practices showed the same trend as in soil gravimetric water content including all investigated soil layers. Previous studies revealed similar results where conservation tillage systems achieved maximum soil water storage compared with conventional tillage systems [20,23]. Jabro et al. [57] drew the similar conclusions and declared that conservation tillage systems increased soil water storage. There is a negative correlation among surface BD with the surface as well as subsurface P, SWC, SWS, and Ks (Table 9). Moreover, a positive correlation was recorded among surface P with surface and sub-surface SWC, SWS whilst negative correlation was observed with surface and subsurface Ks. Surface SWC was positive correlated with surface and subsurface SWS, Ks. In addition, a positive correlation was found among surface SWS with surface and subsurface Ks. Surface Ks was positive corelated with subsurface BD, SWS and Ks whilst negative correlation was found with P and SWC and this coincides with the results noted by [29,53,57].

**Table 9.** Pearson correlation between soil bulk density, porosity, water content, water storage, hydraulic conductivity for tillage measures.

| Soil Depths (cm) | Parameters | Soil Layers (cm) | | | | | | | | | | | | | | |
| --- | --- | --- | --- | --- | --- | --- | --- | --- | --- | --- | --- | --- | --- | --- | --- | --- |
| | | 0–10 | | | | | 10–20 | | | | | 20–40 | | | | |
| | | BD | P | SWC | SWS | Ks | BD | P | SWC | SWS | Ks | BD | P | SWC | SWS | Ks |
| 0–10 | BD | | −1 ** | −0.02 | −0.04 | −0.135 | 0.833 ** | −.833 ** | −0.002 | −0.04 | −0.201 | 0.755 ** | −.758 ** | −0.129 | −0.62 | −0.262 |
| | P | | | 0.22 | 0.05 | −0.152 | −0.833 ** | 0.833 ** | 0.002 | 0.04 | 0.201 | −0.755 ** | 0.758 ** | 0.129 | 0.62 | 0.262 |
| | SWC | | | | 0.526 ** | 0.355 * | 0.09 | −0.09 | 0.611 ** | −0.855 ** | 0.081 | 0.207 | −0.207 | 0.537 ** | −0.254 | −0.222 |
| | SWS | | | | | 0.02 | −0.04 | 0.01 | 0.655 ** | 0.855 * | −0.02 | −0.64 | 0.086 | −0.66 | 0.254 | 0.248 |
| | Ks | | | | | | 0.055 | −0.055 | −0.021 | 0.204 | 0.206 | 0.146 | −0.146 | 0.032 | 0.206 | 0.212 |
| 10–20 | BD | | | | | | | −1 ** | −0.049 | 0.205 | −0.224 | 0.933 ** | 0.934 ** | −0.107 | −0.278 | −0.083 |
| | P | | | | | | | | 0.049 | −0.205 | 0.224 | −0.933 ** | 0.934 ** | 0.107 | 0.278 | 0.083 |
| | SWC | | | | | | | | | 0.526 | −0.012 | 0.009 | −0.009 | 0.338 * | −0.78 | 0.048 |
| | SWS | | | | | | | | | | 0.02 | −0.57 | −0.08 | −0.122 | 0.526 ** | −0.04 |
| | Ks | | | | | | | | | | | −0.187 | 0.187 | 0.122 | 0.09 | −0.122 |
| 20–40 | BD | | | | | | | | | | | | −1 ** | −0.065 | −0.04 | −0.104 |
| | P | | | | | | | | | | | | | 0.065 | 0.04 | −0.104 |
| | SWC | | | | | | | | | | | | | | 0.526 ** | −0.313 |
| | SWS | | | | | | | | | | | | | | | −0.05 |
| | Ks | | | | | | | | | | | | | | | |

Indicates significance at * $p < 0.05$, ** $p < 0.01$.

### 4.5. Influences of Tillage System on Soil Temperature

In soil eco-systems ST is highly correlated with nutrient cycling processes [69]. When it exceeds 10 °C to the highest levels 25 °C to 35 °C, most soil micro-organisms are active. In this research, the daily soil temperature was greater than 5 °C during the growing seasons. Soil temperature is mainly correlated with air temperature and conservation tillage farming plays a significant role in lowering the soil temperature [70]. In agro-eco-system the ST pattern is considered to be associated with variations in solar radiation [71]. The daily radiation changes can affect the ST. Our results depicted that NTS, and CTS decreased ST, because of straw accumulation. A previous study declared that conservation tillage measures decreased ST over CT soil practices [70]. Additionally, the results also suggest the highest soil temperature occurs under CT treatment over NT might be because of surface difference; under no-till farming, the surface of soil is partially shielded by residue remnants from the earlier crops, prompting the soil to absorb less solar radiation [72]. Additionally, under conventional tillage systems the tillage depth makes the soil porous and consequently, the soil probable has lesser thermal conductivity [73]. This variation leads to a maximum retention of heat for conventional tillage systems.

This is consistent with many previous studies all around the world that reported higher ST for CT compared with conservation tillage [29,66,69]. However, our consequences are in contrast with [74] who recorded that straw treated plots had a higher ST. Our findings suggest that NTS, CTS and NT decreased ST. This was mainly due to increased availability of SWC and Ks with the NT, NTS, and CTS techniques [75]. Moreover, conservation tillage increased the infiltration rate which in turn facilitated water movement towards bottom soil layers and decreased ST [75].

### 4.6. Effect of Tillage Systems on Soil TN, AP, TK, C/N Ratio, pH and The Corresponding Z Score

The dry-land agricultural practical pattern of conservation tillage was successfully confirmed to accelerate soil nutrients release, thus increasing the nutrient status of soil and availability to crops; this chiefly occurred through the synergistic regulation of hydro-thermal situation of soil. This pattern possibly a more proficient scenario to substitute CT to conserve and manage fertility and quality of soil in dry-land farming [23,31,33]. Moreover, it improves the soil health indicators; which play a significant role in short- and long-term agriculture sustainability [2,27]. Nevertheless, intensive cropping, continuous ploughing led to serious soil fertility decline which threatens soil and crop sustainability.

The impacts of conservation tillage strategies (e.g., straw-return and straw incorporation) on soil nutrient accumulation and productivity is still a matter of discussion since studies in different soil types and environmental conditions have led to inconsistent findings [33]. Moreover, no-till benefits maybe sustained by several variables including the climatic conditions, soil characteristics and management techniques (e.g., duration of tillage, crop type, and fertilizer application) [37]. Additionally, some aurthors showed NT benefits in long-term studies [2,3,14], For, the recent study, the CTS, NTS, and NT measures significantly increased TN and AP at different soil layers over the three-year study (Table 7). This is consistent with results reported by [18,33,34]. Under CTS, and NTS higher soil TN over CT may be because of microbial biomass and immobilization of nitrogen in straw [76]. These results are in agreement with [18,33,53] who examined the maximum TN accumulation under conservation tillage than CT. Additionally, the highest soil AP in CTS and NTS was attributed to straw application which releases essential soil nutrients [34]. Conversely, the AP with NTS and CTS was not significantly different compared with CT in the subsoil layers; maybe because the straw was incorporated into the top soil layer. This concurs with results reported by [33,77]. Moreover, the NTS technique was expected to increase the TN and AP; nevertheless, the findings rather depicted a reduction trend. The reason might be as a consequence of the slow straw decomposition rate, thereby affecting the TN and AP in soil system [34]. In addition, our results depicted that the implementation of no tillage systems reduced the soil nutrient losses as compared to CT. CT practices in agriculture can mostly lead to a decline in soil nutrients because of destruction of the soil structure and

aggravated soil organic matter decomposition [78]. Overall, the TN and AP concentrations increased in the top soil-layer under the CTS, NTS, and NT techniques. The soil total potassium was not affected with different tillage techniques across different soil layers; this is probably as a consequence of the short-term straw application and perhaps residues need more time for soil total K improvement [79]; this coincides with the results observed by [33,34]. Additionally, the C/N ratio was the highest under conservation tillage treatments compared with CT. Earlier, Sadiq et al. [1] found that NTS, CTS, and NT increased soil organic carbon by 11, 7.3 and 7%, respectively compared to CT. The higher C/N ratio is due to a slower decomposition of the added organic waste [80]. Moreover, in the subsurface soil layer the C/N ratio was lower under NT and NTS which was attributed to a maximum accumulation of TN [81]. Our finding confirms the results achieved by [82] who reported a minimum C/N ratio under no-tilled soil conditions. Jat et al. [53] drew a same conclusion for soil C/N ratio under a NT tillage system. Furthermore, this also concurs with results reported by [18] who reported a lowest C/N ratio under NT farming conditions compared with a reduced tillage system. The pH of the soil was not significantly influenced by tillage practices, though it tended to be lesser in NTS, CTS, and NT compared with CT because of the root exudation and acidifying consequences of organic matter mineralization [83]. The lower value of soil pH is a temporary effect due mainly to the respiratory process of soil microbes and straw decomposition which produce the organic acids which lead to a decreased soil pH [84]. These results are in similar with those of [85] who declared that conservation tillage reduces the soil pH but with a statistically non-significant difference.

The CTS, NTS and NT treatments affected the Z-score of measured soil nutrients in the various soil layers (Table 10). In surface and sub-surface soil layer, NTS, NT and CTS showed the highest scores for the analyzed soil nutrients, for instance TN, AP, TK, C:N ratio, and pH compared with CT. The effect of NT, NTS, and CTS on the improvement of soil nutrients were decreased in the subsurface soil layers which indicated a lower score in comparison with the surface soil layer. Regarding different treatments with different soil layers CTS gives a better score as compared to the rest of the treatments, NTS also showed a better score than NT and CT. Overall, NT, NTS and CTS showed positive scores compared to CT. The CTS, NTS and NT treatments produced higher scores compared with CT, however, the influence of NT decreased with increasing soil depth. From the total Z-score calculation, CTS had a highest score for subsurface soil layers whilst the NTS had the highest score for the surface soil layer in the case of soil pH. The Z-score results demonstrated that, after short-term NT, the soil system did not modify as with CTS and NTS, as suggested by other scientists for instance [1,2,33,86,87].

Principal component analysis (PCA) isolated five principle components according to the Jolliffe cut-off value [50]. In the surface soil layer, the average variance was equal to 70% of the total variance. PC1 contained the maximum loading (35%) and PC2 contained 20% of the maximum loading, whilst the least (15%) loading of the total variance was recorded in PC3 and included the surface TN, AP, K, and C:N ratio (Figure 7). In the sub-surface soil layer, the average variance is equal to 85% of the total variance. PC1 contains the maximum loading (40%) and PC2 contains 25% of the maximum loadings, whilst the least loading (20% of the total variance) was recorded in PC3, which included sub-surface TN, AP, TK, and C:N ratio as shown in Figure 8. The observations plot points of the investigated soil layers drawn by the interaction of PC1/PC2 and PC1/PC3. PC4 and PC5 don't permit one to add additional information which is why they are not plotted. The graphs show that in PC1/PC2 and PC1/PC3 combination the CTS, NTS and NT occupied a more extreme position as compared to CT (Figures 7 and 8). The PCA analysis demonstrates that CTS, NTS, and NT occupied a more specified position in the soil system over CT. Furthermore, it also indicates that the found points of NT was nearer to the central point of PCA components in comparison with NTS and CTS.

**Table 10.** Z score results of soil chemical indicators under tillage systems during 3-year study.

| Treatment | Parameters | | | | | | | | | | | |
|---|---|---|---|---|---|---|---|---|---|---|---|---|
| | Soil Layer | | | | | | | | | | | |
| | 0–10 cm | | | | 10–20 cm | | | | 20–40 cm | | | |
| | Year | | | | | | | | | | | |
| | 2017 | 2018 | 2019 | Total Score | 2017 | 2018 | 2019 | Total Score | 2017 | 2018 | 2019 | Total Score |
| | Soil TN | | | | | | | | | | | |
| CT | −1.64 | −1.40 | −1.66 | −1.56 | −1.56 | −1.64 | −1.43 | −1.57 | −1.45 | −1.27 | −1.44 | −1.38 |
| CTS | 0.98 | 0.57 | 0.92 | 0.82 | 1.03 | 0.62 | 1.38 | 0.98 | 1.30 | 1.43 | 1.12 | 1.28 |
| NT | 0.10 | −0.41 | −0.58 | −0.29 | −0.14 | 0.08 | 0.17 | −0.11 | 0.38 | −0.47 | −0.37 | −0.15 |
| NTS | 0.54 | 1.23 | 0.15 | 0.64 | 0.67 | 0.94 | −0.12 | 0.41 | −0.22 | 0.31 | 0.69 | 0.26 |
| | Soil AP | | | | | | | | | | | |
| CT | −1.29 | −0.88 | −1.40 | −1.19 | −1.36 | −0.89 | −1.27 | −1.23 | −0.88 | −0.85 | −1.28 | −1.00 |
| CTS | 1.49 | 1.63 | 1.29 | 1.47 | 1.21 | 1.66 | 1.40 | 1.39 | 1.68 | 0.22 | 0.33 | 0.74 |
| NT | 0.09 | −0.74 | −0.36 | −0.33 | −0.50 | −0.64 | −0.51 | −0.50 | −0.24 | −0.87 | −0.47 | −0.52 |
| NTS | −0.29 | −0.25 | 0.46 | −0.026 | 0.64 | −0.12 | 0.38 | 0.34 | −0.56 | 1.54 | 1.41 | 0.79 |
| | Soil TK | | | | | | | | | | | |
| CT | −0.18 | −0.42 | 0.06 | −0.18 | −0.22 | −0.15 | −0.27 | −0.14 | −0.05 | −0.13 | 0.08 | −0.03 |
| CTS | 0.48 | 0.65 | 0.29 | 0.47 | 0.58 | 0.72 | 0.69 | 0.57 | 0.82 | 0.50 | 0.49 | 0.60 |
| NT | −0.22 | −0.35 | −0.17 | −0.24 | 0.11 | 0.32 | −0.35 | −0.02 | 0.34 | −0.25 | −0.28 | −0.06 |
| NTS | 0.32 | −0.14 | 0.51 | 0.23 | 0.45 | 0.55 | 0.57 | 0.52 | 0.20 | −0.18 | 0.19 | 0.07 |
| | C/N ratio | | | | | | | | | | | |
| CT | −1.52 | −1.25 | −1.05 | −1.27 | −0.29 | −1.45 | −1.36 | −1.03 | −1.21 | −1.75 | −1.02 | −1.32 |
| CTS | 1.22 | 1.07 | 0.97 | 1.08 | 0.58 | 1.25 | 1.89 | 1.17 | 1.05 | 1.31 | 1.20 | 1.18 |
| NT | 1.41 | 1.13 | 0.88 | 1.14 | 1.50 | 0.87 | 0.27 | 0.88 | 1.85 | 0.76 | 1.74 | 1.45 |
| NTS | 1.24 | 1.40 | 1.02 | 1.22 | 1.70 | 1.43 | 0.82 | 1.24 | 1.42 | 0.17 | 0.98 | 0.85 |
| | pH | | | | | | | | | | | |
| CT | −0.57 | 0.10 | 0.01 | −0.15 | −0.70 | 0.20 | 0.13 | −0.09 | −0.18 | −0.20 | 0.08 | −0.10 |
| CTS | 0.93 | 1.89 | 0.09 | 0.97 | 0.37 | 0.96 | 1.39 | 0.70 | 0.71 | 0.92 | 0.65 | 0.76 |
| NT | −0.34 | 0.90 | 0.20 | 0.25 | −0.45 | 0.08 | 0.60 | 0.10 | −0.40 | 0.03 | 0.67 | 0.1 |
| NTS | 0.98 | 1.52 | 0.50 | 1.00 | 0.87 | 1.22 | 1.04 | 0.90 | 0.80 | 1.03 | 0.94 | 0.92 |

TN = soil total nitrogen; AP = available phosphorous; TK = soil total potassium; C/N = C/N ratio; pH, soil pH. CT: conventional tillage; CTS: conventional tillage with crop straw incorporation; NT: no-tillage; NTS: no-tillage with crop straw-return.

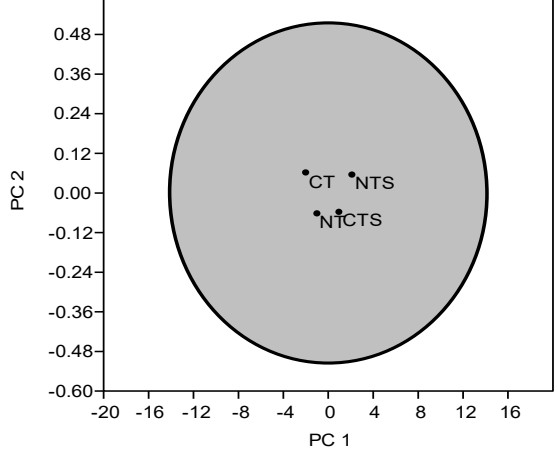

(**a**)

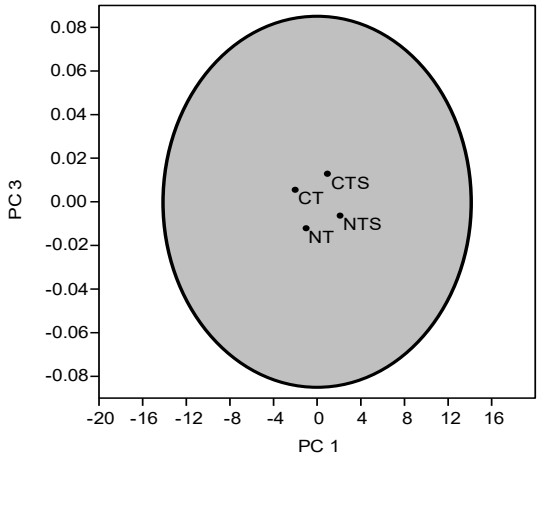

(**b**)

**Figure 7.** Scatter plot of the principle components PC1/PC2 (**a**) and PC1/PC3 (**b**) in the 0–10cm soil layer.

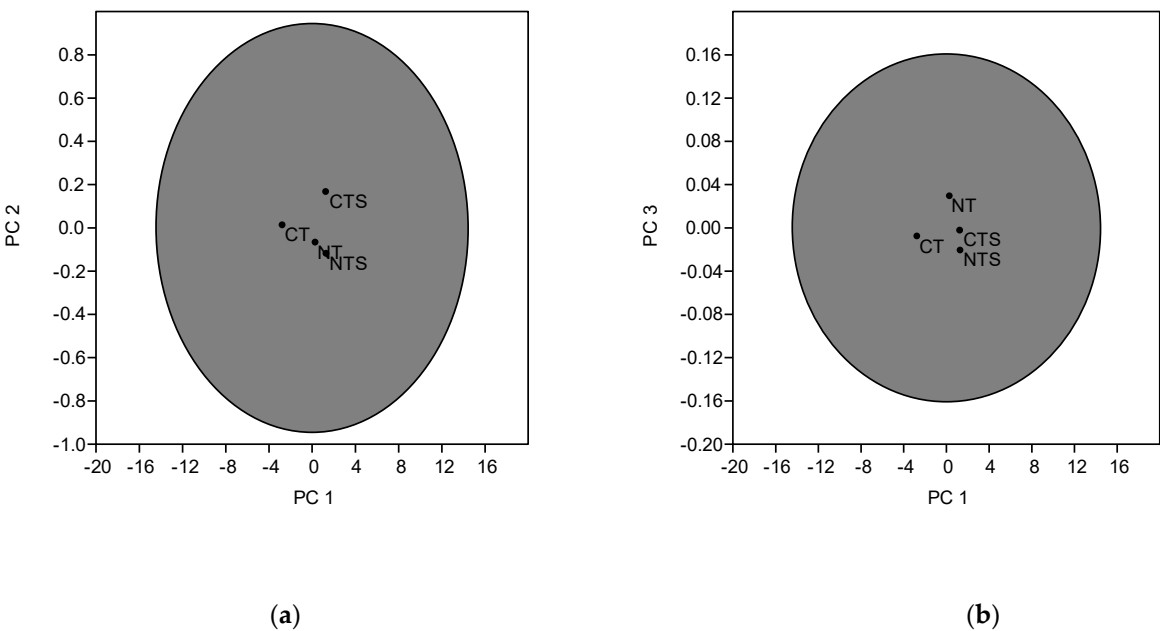

(a)             (b)

**Figure 8.** Scatter plot of the principle components PC1/PC2 (**a**) and PC1/PC3 (**b**) in the 10–40 cm soil layer.

As expected, the PCA shows that the CT was less affected by the surface and sub-surface soil TN, AP, K, and C:N ratio accumulation in the soil system in comparison with CTS, NTS, and NT. Actually, the found points of conventional soil tillage practice were closer compared with conservation tillage measures and closer to the central point of the PCA components (Figures 7 and 8). Moreover, the found points of NT were also nearer to central point of PCA compared with NTS and CTS. No-till treatment was less affected than CTS and NTS included soil TN, AP, TK, and C:N ratio in 0–40 cm soil layer. The outcomes depicted that, after short-term no tillage practice, the soil system did not change compared with NTS and CTS as recommended by other authors such as [18,33]. Additionally, PCA analysis also depicted that general, compared with straw-return to the no-till practice and sole no-till measures, the observation points of straw incorporation with conventional tillage system were far from the PCA components. Accordingly, straw-return to the no-till practice and sole no-till measure did not increase nutrient accumulation compared with straw incorporation with conventional tillage systems, including surface and sub-surface sub-surface TN, AP, TK, and C:N ratio accumulation. Subsequently, the long-term application of straw-return to the no-till practice and sole no-till practice may improve nutrient concentration in the soil system, as suggested by [1,34].

*4.7. Influence of Conservation Tillage System on Crop Agronomic Traits*

Straw implementation and soil inversion elimination are the most valuable techniques for agriculture sustainability. Results from recent studies imply that sustainable crop production can be achieved with the application of CTS, NTS and NT techniques, particularly CTS and NTS because of their soil nutrients retain the ability for appropriate crop utilization, as found in other studies [2,18]. Under NTS, and CTS higher spring wheat agronomic traits than CT was due to the presence of straw, as straw contains organic matter which improves the soil physical and chemical quality indicators contributing to improved crop agronomic traits [34]. The straw retention or incorporation benefits the soil fertility, therefore it improves crop agronomic traits after long-term application as found by [3,14]. We found a significant improvement in the crop agronomic performance after 3-yrs of straw application, which is consistent with the results of [27,33].

In addition, the higher wheat agronomic properties under NT treatment was attributed to maximum soil nutrient accumulation which leads to increased crop production. Moreover, the NT benefits mainly depend on the specific production environment. NT contribution towards crop agronomic traits improvement is perhaps best realized in an environment where the annual precipitation is not more than 300 mm. NT systems may not produce noticeably higher crop production compared to a CT system in adequate rainfall areas [35]. Many scientists have found that long-term implementation of conservation tillage strategies, particularly no-tillage, benefits soil nutrients, thus improving crop productivity [2,14]. We found that NT significantly increased wheat agronomic traits over three years. Nevertheless, there are also other authors who have reported reductions in crop growth and yield under conservation tillage strategies [35,36] due to nitrogen immobilization which resulted in a reduction in crop growth and yield. Moreover, the observed reduction in plant growth and dry matter yield in the first year may have been due to increased temperature and less rainfall as shown in (Supplementary Material Figure S1). The maximum agronomic benefits in the last studied year could have been because the straw need sufficient decomposition time to be effective [87]. Moreover, different conservation tillage strategies improved agronomic traits (seed yield and thousand grain weight) compared with CT. Earlier, Sadiq et al. [1] found that NTS, CTS, and NT increased crop yields by 33, 26, and 18%, respectively, over CT. In this respect, our findings are in line with [23] by indicating the positive effect of conservation tillage on crop growth and yield. These findings are in contrast with [68], who declared that no-till measures significantly decreased the crop yield compared with CT.

*4.8. Soil Health*

Soil health estimation was performed by using various soil physical and chemical quality indicators obtained from different tillage strategies. The measured soil health indicators were BD, P, SWC, SWS, Ks, ST, TN, AP, TK, C/N ratio, and pH. These soil parameters are a consequence of the effects of different tillage techniques on soil health development. Over wide-ranging estimation score consequences obtained from Z-score calculation of the overall research data that we have measured in the three years including soil physicochemical characteristics as soil health indicators; we found that the maximum score for each studied parameter was achieved using conservation tillage strategies (Table 10 and Figure 4) indicating that conservation tillage pattern was the best management method for enhancing soil health. Our study demonstrated that conservation tillage decisively and undoubtedly affects most of the important soil health indicators, including the soil chemical quality indicators that we measured. Previous studies reported that different conservation tillage practices improved soil physico-chemical, hydraulic characteristics and crop yields [4,20,29,64]. Conservation soil tillage technique adoption is the effort to attain soil and crop sustainability under climate change, implementation of conservation tillage farming improved soil function and services and improves soil health [2]. This is consistent with other authors [30,32,33].

**5. Conclusions**

Soil bulk density (and therefore soil porosity) showed non-significant differences in the 0 to 40 cm soil layer for the same year. A small bulk density increase and a soil porosity decrease for the conventional tillage system and straw-return with no-tillage system over other tillage systems was recorded during the study. The results of this study indicated that it is possible to improve the soil hydraulic characteristics and soil water content through conservation tillage farming compared with conventional tillage. However, on the basis of this study's outcome, we conclude that the different tillage methods used in this study did not significantly increase soil bulk density, soil water content, water storage and hydraulic conductivity because parallel pore space was maintained in the sandy texture soil. Normally, sandy texture soils are susceptible to the compaction which leads to unfavorable physical and hydraulic soil characteristics, irrespective of tillage

type. Conservation tillage strategies notably minimized the soil temperature more than the conventional tillage practice. Straw implementation either coupled with conventional tillage or no-tillage systems noticeably improved the soil nutrient accumulation more than conventional soil practices in the 0–40 cm soil layer. Soil pH was not affected by different tillage systems though it tended to be lower in conservation tillage practices compared to soil conventional tillage. We recorded an improvement in crop growth and yield for soil conservation tillage systems compared with conventional tillage. The Z-score formula revealed that the conservation tillage is the most effective approach for increasing soil health. More comprehensive studies are needed concerning specific soil properties to produce further powerful data for determining the health of soil.

**Supplementary Materials:** The following are available online at https://www.mdpi.com/article/10.3390/su13158177/s1.

**Author Contributions:** The research field has been managed by G.L., with ploughing, fertilizing, planting, weeding and harvesting. M.S. and G.L. collected the field and laboratory data. M.S. analyzed data, and wrote the article. G.L., N.R. and M.M.T. reviewed the article. All authors have read and agreed to the published version of the manuscript.

**Funding:** This experimental study was received outside financial support by financial special project of Gansu province (GSCZZ-20160909), Key Talents Project of Gansu Province V (LRYCZ-2020–1) and Key Research and Development program of Gansu Province (20YF8NA135) China.

**Institutional Review Board Statement:** Not applicable.

**Informed Consent Statement:** Not applicable.

**Data Availability Statement:** Not applicable.

**Acknowledgments:** We appreciatively acknowledge the plan of Gansu Province key research and development (18 YF1NA070), Collective innovation team project (2018C-16) of University in Gansu Province China. We would also like to special thank Ijaz Ali at the National Agriculture Research Center Islamabad Pakistan for his technical assistance. We would also like to special thank the editors and the reviewers for their constructive and helpful comments.

**Conflicts of Interest:** The authors declare no conflict of interest. This manuscript entitled sustainable conservation tillage technique for improving soil health by enhancing soil physico-chemical properties under wheat mono-cropping system conditions by Mahran Sadiq et al. has no potential conflict of interest.

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
