# Peer review of "Sustainable Conservation Tillage Technique for Improving Soil Health by Enhancing Soil Physicochemical Quality Indicators under Wheat Mono-Cropping System Conditions"

_sustainability, doi:10.3390/su13158177_

Round 1

Reviewer 1 Report

It was a real pleasure to read the manuscript submitted for review. The issue presented in it (field experiment) is close to my scientific interests. Research of this type is still valid due to the generally positive impact of conservation crops on the soil environment (physico-chemical and biological properties of the soil). It is very valuable to determine the size of the crop yield. The positive effect of conservation tillage on soil properties while stimulating yield size and quality is a valuable prerequisite for agricultural practice.

I must admit that the manuscript under review is very well and carefully prepared. I appreciate the contribution of the authors' work to the implementation of the 3-year field experiment and to the in-depth analysis of soil samples. The obtained results of the research (numerical values ​​of individual examined features) are not controversial. This shows that no mistakes were made in the methodology of the experiment. The results of the analysis of the physical and chemical properties of the soil confirm my positive opinion. In fact, I have no special comments on the individual chapters of the manuscript. The experimental design and research methodology were reliably presented. The description of the research results obtained is clear and clear. The documentation of the results is statistically verified in a way that does not raise my reservations. The documentation of the results is statistically verified in a way that does not raise my reservations. The discussion of the research results is understandable and adequate to the content under consideration. I also evaluate the selection of bibliography sources positively. The presented conclusions from the obtained research are correctly formulated and correspond to the adopted research objective and research hypothesis.

Conservation tillage improved soil hydraulic properties and improved soil nutrient content. This condition of the soil was the "starting point" to improve the yield of spring wheat. The optimistic conclusion from the research is that conservation tillage appears to be an effective way to improve 'soil health'.

Congratulations to the authors of a very good article!

I wish you good ideas for further research.

I recommend this manuscript for printing in Sustainability.

22.04.2021.

Author Response

Hello, respected reviewer, hope you are doing well,. Thank you very much for your help. Please see the attachment.

Reviewer 2 Report

I recommend choosing another journal that focus on agriculture and/or soil. The manuscript focusses on a case study of a single field and investigated 4 different tillage practices for their effects on physicochemical properties. This a typical (and often) investigated issue in soil science and soil tillage research.

Regardless of this point, the present manuscript needs substantial revisions. Here are some important points:

- tillage effects (NT, CT, conservation tillage etc.) are well studied. What is new in this study? Where are the differences to other studies? Please be more precise in the introduction and explain in more detail (i) where is the “knowledge gap” you want to fill and (ii) what is new in your study.

- add more information about the history of the study site. For instance, how was the field used before the experiment? What kind of tillage practice?

- add more information about the different tillage treatments and used machinery (e.g. tillage depth, machinery types)

- add more information on soil sample collection (e.g. how many core samples were taken in each depth?; how do you collect the soil samples for soil chemical analyses?)

- I prefer to use tables instead of figures for Figure 4, 6, 7 (as you did before)

- some results of this investigation are published by Sadiq et al. 2021; add their findings (e.g. SOC, yield information) to the discussion part

- be more critical: the investigation period is three years. Is this period really sufficient to analyse tillage effects? As shown by many studies, the time to analyse and evaluate tillage effects, especially no-tillage practise, need many years. Thus, add more critical thoughts to your manuscript.

- the overall manuscript is quite long for this kind of study. Please try to condense the manuscript a bit.

Author Response

Hello, respected reviewer, hope you are doing well. We hope our efforts have addressed all of the concerns to level required for publication. Thank you very much for your help. We look forward to hearing from you. Please contact us if there are questions or concerns. Thank you very much. Please see the attachment.

Reviewer 3 Report

Authors compare conventional tillage (CT) with no-tillage (NT), no-tillage + straw (NTS) and CT + straw (CTS) applied to wheat monoculture. The subject is not new but it is great interest.The manuscript contains valuable and interesting data that deserve to be published. The methodology is appropriate and the experimental design is magnificent. References are adequate and very up-to-date. Abstract is adequate and reflects well the content of the article. Nevertheless, the work presents serious deficiencies.

Authors name conservation tillage farming to the NT, NTS and CTS treatments. I understand that in the CTS treatment, the straw is buried by the tillage. Conservation tillage implies leaving at least 35% of residues on the surface. Therefore, CTS cannot be considered as a type of conservation agriculture.

Table 4 can be removed. The information provided by this table is not very relevant.

Regarding the soil characteristics, authors only say that it is sandy and refer the reader to a previous work. But in that previous work, only the soil classification is reported: Calcaric Cambisol. Insufficient to characterize the soil. Authors must provide a table with the main characteristics of the soil. At least: texture, BD, pH, EC and organic matter.

Since changes in the soil are slow, to assess the influence of a soil management system, it does not make sense to analyze the data year by year. Data variability it does not allow observe differences between consecutive years. Reduce the results and discussion to the third year (2019), and focus the analysis on the differences detected after 3 years.

The authors' comments on the percentage of increase or decrease, between years or between soil thicknesses, are very confusing and difficult to understand. They should simplify this whole part and focus on commenting on the significant differences found after 3 years under the different treatments.

Figure 5 is not commented on in the text. If it is not relevant then delete it.

They often speak of non-significant differences. The authors forget that there are only differences when they are significant. Remove all mentions of non-significant differences. In this sense, authors must review the whole manuscript and rewrite it taking into account this observation.

Discussion is poor. Great part of the Discussion chapter is a repetition of results, which are little commented and little justified.

Authors often speak of conservation tillage farming (CTF) without to specify whether they refer to NTS, NT, or both. This leads to confusion. Therefore, they must always keep the same nomenclature. Authors must be more specific when referring to the treatments.

I have included numerous comments in a attached pdf file that the authors should see and attend to the comments that are mentioned.

Author Response

Hello, respected reviewer, hope you are doing well. We hope our efforts have addressed all of the concerns to level required for publication. Thank you very much for your help. We look forward to hearing from you. Please contact us if there are questions or concerns. Thank you very much. Please see the attachment

Round 2

Reviewer 2 Report

Dear authors,

please provide the exact line numbers were you have made changes according to the comments.

I could not find the changes in the Introduction section (comment: "Tillage effects (NT, CT, conservation tillage etc.) are well studied. What is new in this study? Where are the differences to other studies? Please be more precise in the introduction and explain in more detail (i) where is the “knowledge gap” you want to fill and (ii) what is new in your study.")

nor could I find the critical reflection in the Discussion section (comment "Be more critical: the investigation period is three years. Is this period really sufficient to analyze tillage effects? As shown by many studies, the time to analyze and evaluate tillage effects, especially no-tillage practice, need many years. Thus, add more critical thoughts to your manuscript.").

Reviewer 3 Report

Authors have made an effort to answer the questions that were asked. Manuscript has noticeably improved. But there are some aspects that still need to be corrected.

  • Line 252: Ks (0.55 m s-1) --> this high Ks is impossible (equivalent to 1980000 mm h-1), when values higher than 250 mm h-1 are considered excessive. Authors should review the units as I indicated in my first review.
  • Data of the Table 5 indicate an enormous hydraulic conductivity, with impossible values. Units may not be good. Authors should review the units as I indicated in my first review.
  • Lines 397-399 “…the universal concept is that an increase in BD and compaction is stereotypically connected with least conductivity”:  Effective macroporosity that determines Ks are the root tubes generated by herbaceous vegetation and the aggregate faces. Ks does not depend on the absolute value of macroporosity as the authors believe, but on the vertical interconnection that this macroporosity presents. And this interconnection is broken by tillage. For this reason, although tillage can increase macroporosity, most of this macroporosity is not effective. However, NT can decrease the total macroporosity value, but it exhibits a much higher vertical interconnection. This justifies that CT presents Ks lower than NT, which is especially notable in the 0-10 cm layer, which is the most affected by soil management (tillage type: CT or NT).

Round 3

Reviewer 2 Report

Accept in present form